# Towards Anytime Classification in Early-Exit Architectures by Enforcing Conditional Monotonicity

**Metod Jazbec**
UvA-Bosch Delta Lab
University of Amsterdam
`m.jazbec@uva.nl`

**James Urquhart Allingham**
University of Cambridge
`jua23@cam.ac.uk`

**Dan Zhang**
Bosch Center for AI &
University of Tübingen
`dan.zhang2@de.bosch.com`

**Eric Nalisnick**
UvA-Bosch Delta Lab
University of Amsterdam
`e.t.nalisnick@uva.nl`

## Abstract

Modern predictive models are often deployed to environments in which computational budgets are dynamic. Anytime algorithms are well-suited to such environments as, at any point during computation, they can output a prediction whose quality is a function of computation time. Early-exit neural networks have garnered attention in the context of anytime computation due to their capability to provide intermediate predictions at various stages throughout the network. However, we demonstrate that current early-exit networks are not directly applicable to anytime settings, as the quality of predictions for individual data points is not guaranteed to improve with longer computation. To address this shortcoming, we propose an elegant post-hoc modification, based on the Product-of-Experts, that encourages an early-exit network to become gradually confident. This gives our deep models the property of *conditional monotonicity* in the prediction quality—an essential stepping stone towards truly anytime predictive modeling using early-exit architectures. Our empirical results on standard image-classification tasks demonstrate that such behaviors can be achieved while preserving competitive accuracy on average.

## 1 Introduction

When deployed in the wild, predictive models are often subject to dynamic constraints on their computation. Consider a vision system for an autonomous vehicle: the same system should perform well in busy urban environments and on deserted rural roads. Yet the former environment often requires quick decision-making, unlike the latter. Hence, it would be beneficial for our models to be allowed to 'think' as long as its environment allows. Deploying models to resource-constrained devices has similar demands. For example, there are roughly $25,000$ different computing configurations of Android devices [Yu et al., 2019], and developers often need to balance poor user experience in low-resource settings with under-performance in high-resource settings.

*Anytime algorithms* can ease, if not resolve, these issues brought about by dynamic constraints on computations. An algorithm is called *anytime* if it can quickly produce an approximate solution and subsequently employ additional resources, if available, to incrementally decrease the approximation error [Dean and Boddy, 1988, Zilberstein, 1996].[1] Anytime computation is especially valuable when resources can fluctuate during test time. Returning to the example of mobile devices, an anytime

---

[1] We give a complete definition in Section 2.

algorithm would be able to gracefully adapt to the computational budget of whatever class of device it finds itself on: low-performing devices will depend on intermediate (but still good) solutions, while high-performing devices can exploit the model's maximum abilities.

*Early-exit neural networks* (EENNs) [Teerapittayanon et al., 2016] have attracted attention as anytime predictors, due to their architecture that can generate predictions at multiple depths [Huang et al., 2018]. However, as we demonstrate, EENNs do not exhibit *anytime* behavior. Specifically, current EENNs do not possess *conditional monotonicity*, meaning that the prediction quality for individual data points can deteriorate with longer computation at test time. This lack of monotonicity leads to a decoupling of computation time and output quality—behavior that is decisively not *anytime*.

In this work, we improve EENNs' ability to perform anytime predictive modeling. We propose a lightweight transformation that strongly encourages monotonicity in EENNs' output probabilities as they evolve with network depth. Our method is based on a Product-of-Experts formulation [Hinton, 1999] and exploits the fact that a product of probability distributions approximates an 'and' operation. Thus by taking the product over all early-exits computed thus-far, the aggregated predictive distribution is gradually refined over time (exits) to have a non-increasing support. This transformation has an inductive bias that encourages *anytime uncertainty*: the first exits produce high-entropy predictive distributions—due to their large support—and the EENN becomes progressively confident at each additional exit (by decreasing the support and reallocating probability).

In our experiments, we demonstrate across a range of vision classification tasks that our method leads to significant improvements in conditional monotonicity in a variety of backbone models *without re-training*. Moreover, our transformation preserves the original model's average accuracy and even improves it in some cases (e.g. CIFAR-100). Thus our work allows the large pool of efficient architectures to be endowed with better anytime properties, all with minimal implementation overhead.

## 2    Background & Setting of Interest

**Data**    Our primary focus is multi-class classification.[2] Let $\mathcal{X} \subseteq \mathbb{R}^d$ denote the feature space, and let $\mathcal{Y}$ denote the label space, which we assume to be a categorical encoding of multiple ($K$) classes. $\boldsymbol{x} \in \mathcal{X}$ denotes a feature vector, and $y \in \mathcal{Y}$ denotes the class (1 of $K$). We assume $\boldsymbol{x}$ and $y$ are realizations of the random variables $\mathbf{x}$ and $\mathbf{y}$ with the (unknown) data distribution $p(\mathbf{x}, \mathbf{y}) = p(\mathbf{y}|\mathbf{x}) \, p(\mathbf{x})$. The training data sample is $\mathcal{D} = \{(\boldsymbol{x}_n, y_n)\}_{n=1}^N$. Our experiments focus on vision tasks in which $\boldsymbol{x}$ is a natural image; in the appendix, we also report results on natural language.

**Early-Exit Neural Networks: Model and Training**    Our model of interest is the *early-exit neural network* [Teerapittayanon et al., 2016] (EENN). For a given data-point $\boldsymbol{x}$, an EENN defines $M$ total models—$p(\mathbf{y} \,|\, \mathbf{x} = \boldsymbol{x}, \, \boldsymbol{\phi}_m, \, \boldsymbol{\theta}_m)$, $m = 1, \ldots, M$, where $\boldsymbol{\phi}_m$ denotes the parameters for the $m$th classifier head and $\boldsymbol{\theta}_m$ are the parameters of the backbone architecture. Each set of parameters $\boldsymbol{\theta}_m$ contains the parameters of all previous models: $\boldsymbol{\theta}_1 \subset \boldsymbol{\theta}_2 \subset \ldots \subset \boldsymbol{\theta}_M$. Due to this nested construction, each model can be thought of as a preliminary 'exit' to the full architecture defined by $\boldsymbol{\theta}_M$. This enables the network's computation to be 'short-circuited' and terminated early at exit $m$, without evaluating later models. Training EENNs is usually done by fitting all exits simultaneously:

$$\ell\left(\boldsymbol{\phi}_1, \ldots, \boldsymbol{\phi}_M, \boldsymbol{\theta}_1, \ldots, \boldsymbol{\theta}_M; \mathcal{D}\right) = -\sum_{n=1}^N \sum_{m=1}^M w_m \cdot \log p(\mathbf{y} = y_n \,|\, \mathbf{x} = \boldsymbol{x}_n, \, \boldsymbol{\phi}_m, \, \boldsymbol{\theta}_m) \quad (1)$$

where $w_m \in \mathbb{R}^+$ are weights that control the degree to which each exit is trained. The weights are most often set to be uniform across all exits. For notational brevity, we will denote $p(\mathbf{y} = y \,|\, \mathbf{x} = \boldsymbol{x}, \, \boldsymbol{\phi}_m, \, \boldsymbol{\theta}_m)$ as $p_m\left(y \,|\, \boldsymbol{x}\right)$ henceforth.

**Setting of Interest**    At test time, there are various ways to exploit the advantages of an EENN. For example, computation could be halted if an input is deemed sufficiently easy such that a weaker model (e.g. $p_1$ or $p_2$) can already make an accurate prediction, thus accelerating inference time. Such *efficient* computation has been the central focus of the EENNs's literature thus far [Matsubara et al., 2023] and is sometimes referred to as *budgeted batch classification* [Huang et al., 2018].

---

[2]See Appendix B.6 for discussion of anytime regression.

However, in this work we focus exclusively on the related, yet distinct, *anytime* setting motivated in the Introduction. We assume computation is terminated when the current environment cannot support computing and evaluating the $(m+1)$th model. These constraints originate solely from the environment, meaning the user or model developer has no control over them. We assume the model will continue to run until it either receives such an external signal to halt or the final—i.e., $M$th—model is computed [Grubb and Bagnell, 2012]. Furthermore, in line with previous literature on anytime algorithms [Huang et al., 2018, Hu et al., 2019], we assume the model sees one data point at a time when deployed, i.e., the batch size is one at test time.

**Anytime Predictive Models**   To have the potential to be called 'anytime,' an algorithm must produce a sequence of intermediate solutions. In the case of classification, an anytime predictive model must hence output a series of probability distributions over $\mathcal{Y}$ with the goal of forming better and better approximations of the true conditional data distribution $p(\mathbf{y}\,|\,\mathbf{x})$. Specifically, we are interested in the evolution of the *conditional error* for a given $\boldsymbol{x} \in \mathcal{X}$, defined here as $\epsilon_t(\boldsymbol{x}) := D\big(p(\mathbf{y}\,|\,\boldsymbol{x}),\, p_t(\mathbf{y}\,|\,\boldsymbol{x})\big)$ where $D$ denotes a suitable distance or divergence (e.g., KL-divergence or total-variation), and $p_t$ represents a model's predictive distribution after computing for time $t \in \mathbb{R}_{\geq 0}$. In addition to conditional, i.e., instance-level, error we can also consider the evolution of a *marginal error* defined as $\bar{\epsilon}_t := \mathbb{E}_{\boldsymbol{x}\sim p(\mathbf{x})}\big[\epsilon_t(\boldsymbol{x})\big]$. For the remainder of this section, we will use $\epsilon_t$ to refer to both versions of the error and revisit their difference in Section 3. Since this paper focuses on anytime EENNs,[3] we use the early-exit index $m$ as the time index $t$.

**Properties of Anytime Models**   Zilberstein [1996] enumerates properties that an algorithm should possess to be *anytime*. Below we summarize Zilberstein [1996]'s points that are relevant to predictive modeling with EENNs, ordering them in a hierarchy of necessity:

1. *Interruptibility*: The most basic requirement is that the model can be stopped and produce an answer at any time. Traditional neural networks do not satisfy interruptibility since they produce a single output after evaluating all layers. However, EENNs do by returning the softmax probabilities for the $m$th model if the $(m+1)$th model has not been computed yet.

2. *Monotonicity*: The quality of the intermediate solutions should not decrease with computation time. Or equivalently, expending more compute time should guarantee performance does not degrade. For EENNs, monotonicity is achieved if the error $\epsilon_m$ is non-increasing across early-exits: $\epsilon_1 \geq \ldots \geq \epsilon_M$.

3. *Consistency*: The quality of the intermediate solutions should be correlated with computation time. For EENNs, consistency is of most concern when the amount of computation varies across sub-models. For example, if computing the $(m+1)$th model requires evaluating $10\times$ more residual blocks than the $m$th model required, then a consistent architecture should produce a roughly $10\times$ better solution ($\epsilon_{m+1} \approx \epsilon_m/10$).

4. *Diminishing Returns*: The difference in quality between consecutive solutions is gradually decreasing, with the largest jumps occurring early, e.g. $(\epsilon_1 - \epsilon_2) \gg (\epsilon_{M-1} - \epsilon_M)$. Monotonicity and consistency need to be satisfied in order for this stronger property to be guaranteed.

For a traditional EENN (trained with Equation 1), property #1 is satisfied by construction (by the presence of early-exits). Property #2 is not explicitly encouraged by the typical EENN architecture or training objective. Hence, we begin the next section by investigating if current EENNs demonstrate any signs of monotonicity. Properties #3 and #4 are left to future work since, as we will demonstrate, achieving property #2 is non-trivial and is thus the focus of the remainder of this paper.

**Estimating Anytime Properties**   The error sequence $\{\epsilon_m\}_{m=1}^M$ is essential for studying the anytime properties of EENNs. We can equivalently consider the evolution of *prediction quality* measures $\gamma_m$ and aim for a model that exhibits increasing quality, i.e., $\gamma_m \leq \gamma_{m+1}, \forall m$. For the remainder of the paper, we will work with the notion of prediction quality unless otherwise stated. Observe that each $\gamma_m$ (and equivalently, $\epsilon_m$) is always unknown, as it depends on the unknown data distribution $p(\mathbf{y}\,|\,\mathbf{x})$. Consequently, we must rely on estimates $\hat{\gamma}_m$. In the presence of a labeled hold-out dataset, we can use ground-truth labels $y^* \in \mathcal{Y}$ and consider the correctness of the prediction as our estimator:

---

[3]While EENNs are the focus of our work, they are not the only model class that can be used for anytime prediction. In theory, any model that yields a sequence of predictions could be considered.

$\hat{\gamma}_m^c(\boldsymbol{x}) := [\arg\max_y p_m(y \mid \boldsymbol{x}) = y^*]$ where $[\cdot]$ is the Iverson bracket.[4] However, such an estimator provides a crude signal (0 or 1) on a conditional level (per data point). As such, it is not congruent with conditional monotonicity or consistency. Due to this, we focus primarily on the probability of the ground-truth class $\hat{\gamma}_m^p(\boldsymbol{x}) := p_m(y^* \mid \boldsymbol{x})$ when examining anytime properties at the conditional level. Finally, estimating marginal prediction quality (error) requires access to $p(\mathbf{x})$. Following common practice, we approximate $\mathbb{E}_{\boldsymbol{x} \sim p(\mathbf{x})}$ by averaging over the hold-out dataset. Averaging $\hat{\gamma}_m^c(\boldsymbol{x})$ and $\hat{\gamma}_m^p(\boldsymbol{x})$ corresponds to the test accuracy and the average ground-truth probability at the $m$th exit, respectively.

## 3 Checking for Monotonicity in Early-Exit Neural Networks

In this section, we take a closer look at *state-of-the-art* early-exit neural networks and their monotonicity, a key property of anytime models. Our results show that EENNs are marginally monotone but lack conditional monotonicity in the prediction quality. For ease of presentation, we focus here on a particular EENN (*Multi-Scale Dense Net* (MSDNet); Huang et al. [2018]) and dataset (CIFAR-100). Section 6 shows that this observation generalizes across other EENNs and datasets.

**Marginal Monotonicity**     The left plot of Figure 1 reports MSDNet's test accuracy (**blue**) on CIFAR-100 as a function of the early exits. Accuracy has been the primary focus in the anytime EENN literature [Huang et al., 2018, Hu et al., 2019], and as expected, it increases at each early exit $m$. Thus MSDNet clearly exhibits *marginal monotonicity*: monotonicity on average over the test set. The same figure also shows that the average ground-truth probability (**orange**) increases at each $m$, meaning that MSDNet demonstrates marginal monotonicity in this related quantity as well.

**Conditional Monotonicity**     Yet for real-time decision-making systems, such as autonomous driving, it is usually not enough to have marginal monotonicity. Rather, we want *conditional monotonicity*: that our predictive model will improve instep with computation *for every input*. If conditional monotonicity is achieved, then marginal monotonicity is also guaranteed. However, the converse is not true, making conditional monotonicity the stronger property.

To check if MSDNet demonstrates conditional monotonicity, examine the middle plot of Figure 1. Here we plot the probability of the true label, i.e., $\hat{\gamma}_m^p(\boldsymbol{x})$, as a function of the early exits for 10 randomly sampled test points (again from CIFAR-100). We see that—despite MSDNet being marginally monotonic—it is quite unstable in the probabilities it produces per exit. For example, the **purple** line exhibits confidence in the correct answer ($\sim 75\%$) at the second exit, drops to near zero by the fourth exit, becomes confident again ($\sim 80\%$) at exits 5 and 6, and finally falls to near zero again at the last exit. Thus MSDNet clearly does not exhibit the stronger form of monotonicity.

We next quantify the degree to which MSDNet is violating conditional monotonicity. First, we define the *maximum decrease* in the ground-truth probability as $\mathrm{MPD}^*(\boldsymbol{x}) := \max_{m' > m} \left\{ \max(\hat{\gamma}_m^p(\boldsymbol{x}) - \hat{\gamma}_{m'}^p(\boldsymbol{x}), 0) \right\}$. For various thresholds $\tau \in [0, 1]$, we count how many test instances exhibit a probability decrease that exceeds the given threshold: $N_\tau := \sum_n [\mathrm{MPD}^*(\boldsymbol{x}_n) \geq \tau]$. The results are depicted in the right panel of Figure 1. We see that for $\sim 25\%$ of test points, the ground-truth probability decreases by at least $0.5$ at some exit. In a separate analysis (see Appendix B.1.2), when examining the correctness of prediction measure $\hat{\gamma}_m^c(\boldsymbol{x})$, we find that $\sim 30\%$ of test examples exhibit decreasing trajectories across early exits.[5] A similar finding was reported by Kaya et al. [2019]. Both observations demonstrate that MSDNet, a state-of-the-art EENN, does not exhibit conditional monotonicity nor is it close to achieving it.

## 4 Transformations for Conditional Monotonicity

In light of the aforementioned challenges with current EENNs in the anytime setting, we explore potential lightweight modifications that could encourage, or even enforce, conditional monotonicity. We first suggest a strong baseline for encouraging monotonicity, taking inspiration from Zilberstein [1996]. We then introduce our primary methodology, which is based on product ensembles.

---

[4]The Iverson bracket $[\texttt{cond}]$ is 1 if $\texttt{cond}$ is true and 0 otherwise.

[5]This implies that the model incorrectly classifies the data point after correctly predicting it at some of the earlier exits: $\hat{\gamma}_m^c(\boldsymbol{x}) = 1$ and $\hat{\gamma}_{m'}^c(\boldsymbol{x}) = 0$ for some $m' > m$.

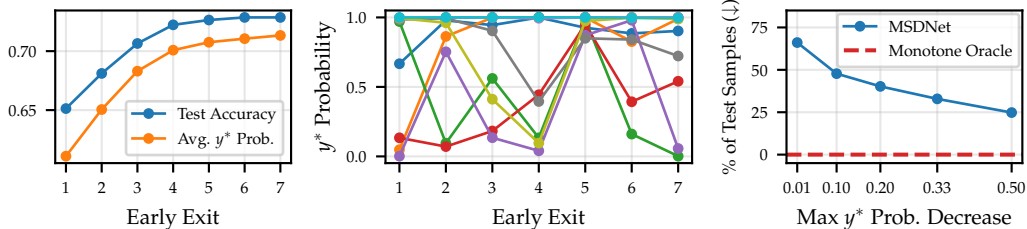

Figure 1: Multi-Scale Dense Network (MSDNet; Huang et al. [2018]) results on the test set of CIFAR-100. *Left*: test accuracy and average ground-truth probability at each early exit, the two measures of marginal monotonicity we study in this work. *Middle*: ground-truth probability trajectories (i.e., $\{\hat{\gamma}_m^p(\boldsymbol{x})\}_{m=1}^M$) for 10 random test data points. *Right*: aggregation of analysis on conditional monotonicity for MSDNet. Specifically, we compute the maximum decrease in ground-truth probability for each test data point and then plot the percentage of test examples where the drop exceeds various thresholds. A **red** dashed line denotes the performance of an oracle model with perfect conditional monotonicity, i.e., none of the data points exhibit any decrease in the ground-truth probability.

## 4.1 Caching Anytime Predictors

As noted by Zilberstein [1996], conditional monotonicity can be encouraged by caching the best prediction made so far and returning it (instead of the latest prediction). Yet this requires the ability to measure the prediction quality of intermediate results. While previously introduced quantities, such as ground-truth probability $\hat{\gamma}_m^p(\boldsymbol{x})$, can serve to examine anytime properties before deploying the model, their reliance on the true label $y^*$ renders them unsuitable for measuring prediction quality at test time. As a proxy, we propose using the EENN's internal measure of confidence. We found that simply using the softmax confidence performs well in practice. At an exit $m$, we compute $C(p_m, \boldsymbol{x}) := \max_y p_m(y|\boldsymbol{x})$ and overwrite the cache if $C(p_j, \boldsymbol{x}) < C(p_m, \boldsymbol{x})$, where $j$ is the exit index of the currently cached prediction. We call this approach *Caching Anytime* (CA) and consider it as a (strong) baseline. While CA is rather straightforward, it has been thus far overlooked in the anytime EENN literature, as the standard approach is to return the latest prediction [Huang et al., 2018, Hu et al., 2019]. Note that if the anytime model displays conditional monotonicity to begin with, applying CA has no effect since the most recent prediction is always at least as good as the cached one.

## 4.2 Product Anytime Predictors

Our proposed method for achieving conditional monotonicity in EENNs is motivated by the following idealized construction.

**Guaranteed Conditional Monotonicity via Hard Product-of-Experts** Consider an EENN whose sub-classifiers produce 'hard' one-vs-rest probabilities: each class has either probability 0 or 1 (before normalization). Now define the predictive distribution at the $m$th exit to be a Product-of-Experts [Hinton, 1999] (PoE) ensemble involving the current and all preceding exits:

$$p_{\Pi,m}(y \mid \boldsymbol{x}) := \frac{1}{Z_m} \prod_{i=1}^{m} \left[ f_i(\boldsymbol{x})_y > b \right]^{w_i}, \quad Z_m = \sum_{y' \in \{1,\ldots,K\}} \prod_{i=1}^{m} \left[ f_i(\boldsymbol{x})_{y'} > b \right]^{w_i} \quad (2)$$

where $b$ is a threshold parameter, $[\cdot]$ is an Iverson bracket, $f_i(\boldsymbol{x}) \in \mathbb{R}^K$ represents a vector of logits at $i$th exit, and $w_i \in \mathbb{R}^+$ are ensemble weights. Assuming that a class $y$ is in the support of the final classifier for a given $\boldsymbol{x}$, i.e. $p_{\Pi,M}(y \mid \boldsymbol{x}) > 0$, then such a model is *guaranteed* to be monotone in the probability of $y$:

$$p_{\Pi,m}(y \mid \boldsymbol{x}) \le p_{\Pi,m+1}(y \mid \boldsymbol{x}), \ \forall\, m . \quad (3)$$

The formal proof can be found in Appendix A. Consequently, if the true class $y^*$ is in the final support, an EENN with this construction is conditionally monotonic in the ground-truth probability $\hat{\gamma}_m^p(\boldsymbol{x})$ (as defined in Section 2).

The intuition behind this construction is that at each exit, the newly computed ensemble member casts a binary vote for each class. If a class has received a vote at every exit thus far, then it remains in the support of $p_{\Pi,m}(\mathbf{y} \mid \boldsymbol{x})$. Otherwise, the label drops out of the support and never re-enters. At each

exit, the set of candidate labels can only either remain the same or be reduced, thus concentrating probability in a non-increasing subset of classes. This construction also has an *underconfidence bias*: the probability of all labels starts small due to many labels being in the support at the earliest exits. Consequently, the model is encouraged to exhibit high uncertainty early on and then to become progressively confident at subsequent exits.

**A Practical Relaxation using ReLU**    Although the above construction provides perfect conditional monotonicity (for labels in the final support), it often results in a considerable decrease in marginal accuracy (see Appendix B.2). This is not surprising since the $0 - 1$ probabilities result in 'blunt' predictions. At every exit, all in-support labels have equal probability, and thus the model must select a label from the support at random in order to generate a prediction. The support is often large, especially at the early exits, and in turn the model devolves to random guessing.

To overcome this limitation, first notice that using $0 - 1$ probabilities is equivalent to using a Heaviside function to map logits to probabilities.[6] In turn, we propose to relax the parameterization, replacing the Heaviside with the ReLU activation function. The ReLU has the nice property that it preserves the rank of the 'surviving' logits while still 'nullifying' classes with low logits. Yet this relaxation comes at a price: using ReLUs allows conditional monotonicity to be violated. However, as we will show in Section 6, the violations are rare and well worth the improvements in accuracy. Moreover, the number of violations can be controlled by applying an upper limit to the logits, as detailed in Appendix B.8.

Therefore our final ***Product Anytime*** (PA) method for conditional monotonicity is written as:

$$p_{\mathrm{PA},m}\left(y \,|\, \boldsymbol{x}\right) := \frac{1}{Z_m} \prod_{i=1}^{m} \max\left(f_i(\boldsymbol{x})_y,\, 0\right)^{w_i}, \quad Z_m = \sum_{y' \in \{1,\ldots,K\}} \prod_{i=1}^{m} \max\left(f_i(\boldsymbol{x})_{y'},\, 0\right)^{w_i} \quad (4)$$

where $\max(\cdot, \cdot)$ returns the maximum of the two arguments. We set the ensemble weights to $w_i = i/M$ since we expect later experts to be stronger predictors. Note that PA's predictive distribution may degenerate into a *zero* distribution such that $p_{\mathrm{PA},m}\left(y \,|\, \boldsymbol{x}\right) = 0, \ \forall\, y \in \mathcal{Y}$. This occurs when the ensemble's experts have non-overlapping support and are thus unable to arrive at a consensus prediction. If a prediction is required, we fall back on the softmax predictive distribution based on the logits of the classifier at the current exit.

**Learning and Post-Hoc Application**    Equation 4 can be used for training an EENN directly via the negative log-likelihood. However, one can also take pre-trained EENNs (e.g., as open-sourced by the original authors) and apply Equation 4 *post-hoc*. For this post-hoc application, the EENN's logits are input into Equation 4 rather than into the usual softmax transformation. As we will demonstrate in the experiments, post-hoc application endows the model with conditional monotonicity without any substantial degradation to the original model's accuracy. In cases such as CIFAR-100, we find that our method actually *improves* the accuracy of the pre-trained model. The only metric we find to be sensitive to training from scratch vs

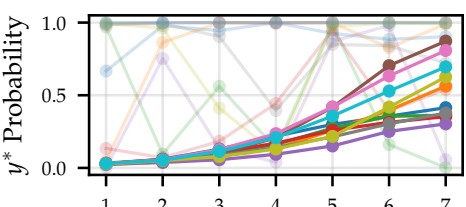

Figure 2: Ground-truth probability trajectories (i.e., $\{\hat{\gamma}_m^p(\boldsymbol{x})\}_{m=1}^{M}$) for 10 random test data points after applying our PA method. For comparison, MSD-Net's trajectories from Figure 1 are reproduced in the background.

post-hoc application is confidence calibration, with the former providing better calibration properties. This result will be demonstrated in Figure 6. Thus, we currently recommend using our method via post-hoc application, as it requires minimal implementation overhead (about three lines of code) and allows the user to easily toggle between anytime and traditional behavior.

For a visualization of how our post-hoc PA affects probability trajectories, see Figure 2. PA manages to rectify the backbone EENN's non-monotone behavior to a large extent. As a concrete example of how this can help with anytime predictions, we can consider again the scenario of Android

---

[6]In the most widely used softmax parameterization of the categorical likelihood, the exponential function is used when transforming logits into probabilities.

phones introduced in Section 1. Deploying the original MSDNet might result in certain data points receiving inferior predictions on a higher-spec device compared to a device with limited computational capabilities. However, if we apply our PA to an MSDNet, such inconsistencies become far less likely due to the model's monotonic behavior.

## 5 Related work

*Anytime Algorithms* [Dean and Boddy, 1988, Zilberstein, 1996] have long been studied outside the context of neural networks. For instance, Wellman and Liu [1994] considered Bayesian networks in anytime settings, while Grubb and Bagnell [2012] proposed an anytime Ada-Boost algorithm [Bishop, 2007] and also studied Haar-like features [Viola and Jones, 2001] for anytime object detection. More recently, modern neural networks have been used in models with only *marginal* anytime properties [Huang et al., 2018, Hu et al., 2019, Liu et al., 2022]. Our research improves upon this work by bringing these models a step closer to having per-data-point anytime prediction, a much stronger form of anytime computation.

*Early-Exit Neural Networks* [Teerapittayanon et al., 2016, Yang et al., 2020, Han et al., 2022a] have been investigated for applications in computer vision [Huang et al., 2018, Kaya et al., 2019] and natural language processing [Schwartz et al., 2020, Schuster et al., 2021, Xu and McAuley, 2023]. A wide variety of backbone architectures has been explored, encompassing recurrent neural networks [Graves, 2016], convolutional networks [Huang et al., 2018], and transformers [Wang et al., 2021]. Furthermore, ensembling of EENNs intermediate predictions has been considered [Qendro et al., 2021, Wolczyk et al., 2021, Meronen et al., 2023], primarily to enhance their uncertainty quantification capabilities. Our objective is to leverage the recent advancements in early-exit networks and make them suitable for true (conditional) anytime predictive modeling.

Related to our critique of current EENNs, Kaya et al. [2019] and Wolczyk et al. [2021] also observed that prediction quality can decline at later exits, calling the phenomenon 'overthinking.' These works address the problem via methods for early termination. Our work aims to impose monotonicity on the ground-truth probabilities, which eliminates 'overthinking' by definition. We further examine the connection to Kaya et al. [2019] and Wolczyk et al. [2021] in Appendix B.5.

*Product-of-Experts* [Hinton, 1999, 2002] ensembles have been extensively studied in the generative modeling domain due to their ability to yield sharper distributions compared to Mixture-of-Experts ensembles [Wu and Goodman, 2018, Huang et al., 2022]. However, in the supervised setting, PoE ensembles have received relatively little attention [Pignat et al., 2022]. PoE is often applied only post-hoc, likely due to training difficulties [Cao and Fleet, 2014, Watson and Morimoto, 2022].

## 6 Experiments

We conduct two sets of experiments.[7] First, in Section 6.1, we verify that our method (PA) maintains strong average performance while significantly improving conditional monotonicity in state-of-the-art EENNs making them more suitable for the anytime prediction task. To check that our findings generalize across data modalities, we also conduct NLP experiments and present the results in Appendix B.4. In the second set of experiments, detailed in Section 6.2, we move beyond prediction quality and shift focus to monotonicity's effect on uncertainty quantification in the anytime setting. Lastly, we point out some limitations in Section 6.3.

**Datasets** We consider CIFAR-10, CIFAR-100 [Krizhevsky et al., 2009], and ILSVRC 2012 (ImageNet; Deng et al. [2009]). The CIFAR datasets each contain 50k training and 10k test images, while ImageNet is a larger dataset with 1.2M training and 50k test instances. The three datasets consist of 10, 100, and 1000 classes, respectively.

**Baselines** We use four state-of-the-art EENNs as baselines and backbone architectures. The first is the Multi-Scale Dense Network (MSDNet; Huang et al. [2018]), which consists of stacked convolutional blocks. To maximize performance at each early-exit, the authors introduce: (1) feature maps on multiple scales, ensuring that coarse-level features are available all throughout the network,

---

[7]Our code is publicly available at `https://github.com/metodj/AnytimeClassification`.

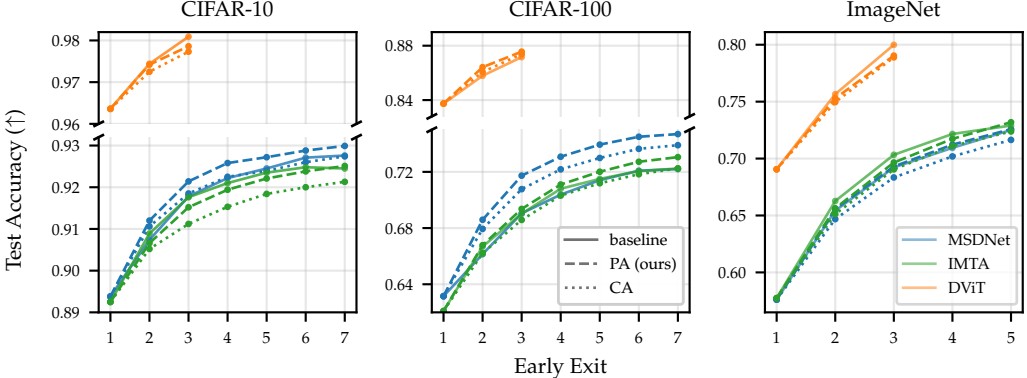

Figure 3: Test accuracy on the CIFAR and ImageNet datasets. Our PA method maintains a competitive performance compared to the baseline models. For each model and dataset, we use the same number of early exits as proposed by the authors. We plot the average accuracy from $n = 3$ independent runs. To enhance the readability of the plot, we omit the error bars here as well as the results of the L2W model and report them in Appendix B.11.

and (2) dense connectivity to prevent earlier classifiers from negatively impacting later ones. The second baseline is the IMTA network [Li et al., 2019], which shares MSDNet's architecture but is trained with regularization that improves collaboration across the classifiers of each exit. The third baseline is the L2W model [Han et al., 2022b], which enhances MSDNet by learning to weight easy and hard samples differently at each exit during training. The intuition is that easier samples should contribute more to the fitting of earlier exits, while harder samples should be more influential for later exits. Lastly, we consider the Dynamic Vision Transformer (DViT; Wang et al. [2021]), which is composed of stacked transformer networks that differ in the number of patches into which they divide an input image. We report results for the T2T-ViT-12 instantiation of DViT. During testing, all baselines employ a softmax parameterization for their categorical likelihood and solely rely on the most recent prediction at each early exit.

## 6.1 Anytime Prediction with Conditional Monotonicity

**Test Accuracy** We first verify that post-hoc application of our PA transformation preserves the original model's test accuracy. Maintaining test accuracy is essential, as simply examining conditional monotonicity in isolation does not provide a complete picture. For instance, a model that predicts a ground-truth probability of zero at every exit would exhibit perfect conditional monotonicity while being useless in practice. Figure 3 reports accuracy per exit for the three backbones models, comparing the original model (—) to our proposals PA (- -) and CA (⋯). We find that our monotonicity methods do not significantly degrade test accuracy in any case. In some instances, such as the MSDNet results on CIFAR-100 (*middle*, **blue**), our PA even improves accuracy. Comparing our proposals, PA tends to outperform CA in accuracy, although the advantage is often slight.

**Conditional Monotonicity** Next, we quantify the degree to which each model is conditionally monotonic, again using the maximum probability decrease metric (MPD*) used in Figure 1 (*right*). Figure 4 reports the percentage of test cases that exhibit a drop in the ground-truth probability at any exit. Firstly, we see that CA and PA drastically improve monotonicity in all cases. Moreover, PA is clearly superior to CA for CIFAR-100 and ImageNet. On CIFAR-10, PA and CA perform on par for smaller decrease thresholds, with PA outperforming CA again for larger thresholds. Notably, on ImageNet, our PA method does not exhibit a drop in ground-truth probability larger than $0.01$ for any test example for all three models.

From the results in Figures 3 and 4, one might wonder if a simpler change to the backbone EENN could lead to similar improvements in conditional monotonicity as our PA. To test this, we looked at 'vanilla' early-exit ensembles where predictive distributions from different exits are combined using either mixture-of-experts (MoE) or product-of-experts (PoE), see Appendix B.2 for the results. While using ensembles is beneficial, it is insufficient to achieve improvements in monotonicity similar to those of PA. This highlights the importance of the activation function (that maps logits

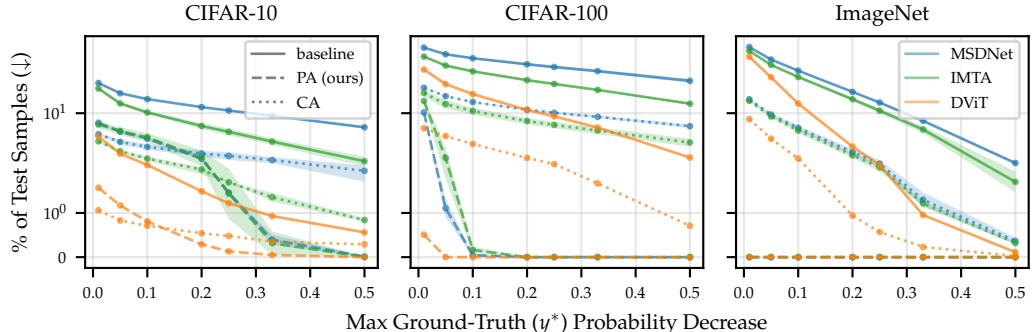

Figure 4: The % of test examples with a ground-truth probability drop exceeding a particular threshold. Our PA significantly improves the conditional monotonicity of ground-truth probabilities across various datasets and backbone models. To better illustrate the different behavior of various methods, a log scale is used for the y-axis. We report average monotonicity together with one standard deviation based on $n = 3$ independent runs. For DViT [Wang et al., 2021], we were unable to perform multiple independent runs, and report the results for various models instantiations instead, see Appendix B.11. Note that for PA, the monotonicity curves on ImageNet (almost perfectly) overlap at $y = 0$ for all backbone models considered. We also report results for the L2W model in Appendix B.11, which are largely the same as for MSDNet.

to probabilities) when it comes to achieving more monotone behavior. We additionally explored if calibrating the backbone EENN would be enough to make the underlying model monotone. While post-hoc calibrating every exit does help with monotonicity, it does not perform as well as PA; see Appendix B.9.

**Monotonicity in Correctness**   Our PA approach yields close to perfect conditional monotonicity when considering probability as the estimator of prediction quality. Yet it is also worth checking if other estimators are monotonic, such as the correctness of the prediction, i.e., $\hat{\gamma}_m^c(\boldsymbol{x})$. For correctness, we observe similar improvements in conditional monotonicity; see Appendix B.1.2 for a detailed analysis. For example, on CIFAR-100, MSDNet 'forgets' the correct answer in $\sim 30\%$ of test cases, whereas applying post-hoc PA reduces these monotonicity violations to $\sim 13\%$ of cases.

## 6.2   Anytime Uncertainty

We also examine the uncertainty quantification abilities of these anytime models. Again consider Figure 2, the probability trajectories obtained from post-hoc PA. The probabilities start low at first few exits, indicating high uncertainty in the early stages of anytime evaluation. The left plot of Figure 5 further supports this insight, showing that the average entropy of the PA predictive distributions is high and reduces at each early exit. Lastly, in the right plot, we present the average size of *conformal sets*[8] at each exit. The set size reflects the model's predictive uncertainty, with larger sets indicating more uncertainty. Thus we see that PA is best at uncertainty quantification: it is quite uncertain at the earliest exits (set size of $\sim 11$) but grows twice as confident by the last exit (set size of $\sim 4$). CA and regular MSDNet keep a relatively constant set size by comparison, only varying between

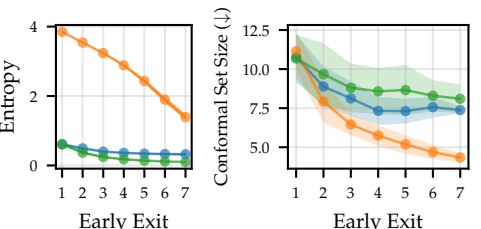

Figure 5: Uncertainty measures for the MSDNet model on the CIFAR-100. *Left*: average test entropy of (categorical) predictive distributions for each early exit. *Right*: mean size of conformal sets at each exit point. To obtain conformal scores, we used Regularized Adaptive Predictive Sets algorithm [RAPS; Angelopoulos et al., 2021]. 20% of test examples were used as a hold-out set to calculate conformal quantiles at each early exit.

---

[8]Conformal sets are prediction sets over $\mathcal{Y}$ that are marginally guaranteed to contain the ground-truth label (without making any assumptions on the underlying data distribution). For more on conformal sets, see Shafer and Vovk [2008].

11 and 7.5. Hence, the monotonicity properties of PA result in an appealing link between uncertainty and computation time; and we term this property as *anytime uncertainty*. This is an appropriate inductive bias for anytime predictive models since, intuitively, making quicker decisions should entail a higher level of uncertainty. Lastly, in Appendix B.7, we present results of conditional level experiments where we find that our PA yields significantly more monotone (i.e., non-increasing) uncertainty patterns.

### 6.3 Limitations

**Label Sets with Small Cardinality** We observe that PA's performance in terms of conditional monotonicity improves as the number of classes $|\mathcal{Y}|$ increases. For instance, PA yields fully monotone trajectories on ImageNet, but not for datasets with smaller number of classes like CIFAR-10, where we detect small violations. This finding is further supported by our NLP experiments in Appendix B.4, where $|\mathcal{Y}|$ is smaller compared to the image classification datasets considered in this section. Based on this finding, we recommend using the CA approach over PA for scenarios in which $|\mathcal{Y}| < 5$.

**Calibration Gap** Due to its aforementioned underconfidence bias, post-hoc PA generates poorly calibrated probabilities during the initial exits, as illustrated in the left plot of Figure 6. Yet this is primarily observed in *expected calibration error* (ECE) and related metrics. It disappears when considering some alternative metrics to ECE, as shown in the right plot of Figure 12. Finetuning with PA as an optimization objective, as opposed to applying it post-hoc, closes the calibration gap in most cases, albeit at some cost to accuracy and monotonicity. See Figure 6 (*right*) for concrete examples of how PA with finetuning affects ground-truth probability trajectories. In addition, we experimented with adaptive thresholding in PA, using different thresholds $b$ for distinct data points, rather than employing 0 for all data points as with ReLU, and found that it can also be beneficial in terms of calibration. For more details on finetuning and adaptive thresholding, as well as a discussion on the relevance of ECE-like metrics in the anytime setting, see Appendix B.3.

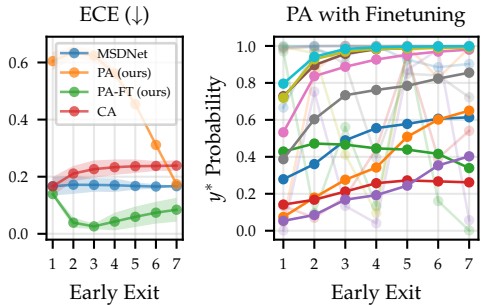

Figure 6: Calibration analysis for MSDNet on CIFAR-100. *Left*: expected calibration error (ECE) during anytime evaluation. *Right*: Ground-truth probability trajectories for 10 random test data points after applying our PA with finetuning (PA-FT) method. For comparison, MSDNet's trajectories from Figure 1 are reproduced in the background.

## 7 Conclusion

We have demonstrated that expending more time during prediction in current early-exit neural networks can lead to reduced prediction quality, rendering these models unsuitable for direct application in anytime settings. To address this issue, we proposed a post-hoc modification based on Product-of-Experts and showed through empirical and theoretical analysis that it greatly improves conditional monotonicity with respect to prediction quality over time while preserving competitive overall performance.

In future work, we aim to further improve EENNs by imbuing them with other critical anytime properties, such as diminishing returns and consistency. Moreover, it would be interesting to study the impact of monotonicity in other settings, such as budgeted batch classification. We report some preliminary results in Appendix B.10, finding the monotonicity to be less beneficial there compared to the anytime setting, which was the focus of our work. We also plan to delve deeper into the concept of anytime uncertainty. Although we have (briefly) touched upon it in this study, we believe a more comprehensive and formal definition is necessary for the development of robust and reliable anytime models.

## Acknowledgments and Disclosure of Funding

We thank Alexander Timans and Rajeev Verma for helpful discussions. We are also grateful to the anonymous reviewers who helped us improve our work with their constructive feedback. MJ and EN are generously supported by the Bosch Center for Artificial Intelligence. JUA acknowledges funding from the EPSRC, the Michael E. Fisher Studentship in Machine Learning, and the Qualcomm Innovation Fellowship.

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

The Appendix is structured as follows:

- We provide a proof of conditional guarantees in EENNs for (hard) PoE in Appendix A.
- We present conditional monotonicity results using alternative estimators of performance quality $\gamma_m$ in Appendix B.1.
- We conduct an ablation study for our PA model in Appendix B.2.
- We propose two modifications to our PA that help with reducing the calibration gap in Appendix B.3.
- We report results of NLP experiments in Appendix B.4.
- We delve into the connections between conditional monotonicity and previously introduced concepts in EENNs literature in Appendix B.5.
- We discuss anytime regression and deep ensembles in Appendix B.6.
- We present conditional monotonicity results using uncertainty measures in Appendix B.7.
- We propose a technique for controlling the violations of conditional monotonicity in PA in Appendix B.8.
- We study the impact of calibration on the conditional monotonicity in Appendix B.9.
- We provide budgeted batch classification results in Appendix B.10.
- We provide additional plots in Appendix B.11.

# A   Theoretical Results

In the following, we provide a proof for conditional monotonicity of (hard) Product-of-Experts ensembles presented in Section 4.2.

**Proposition.** *Let $p_{\Pi,m}\left(\mathbf{y}\,|\,\boldsymbol{x}\right)$ represent the Product-of-Experts ensemble of the first $m$ early-exits. Furthermore, let $y \in \mathcal{Y}$ denote any class that is present in the support of the full-NN product distribution, i.e., $p_{\Pi,M}\left(y\,|\,\boldsymbol{x}\right) > 0$. Under the assumption that the Heaviside activation function is used to map logits to probabilities in each $p_m\left(y\,|\,\boldsymbol{x}\right)$, it follows that:*

$$p_{\Pi,m}\left(y\,|\,\boldsymbol{x}\right) \leq p_{\Pi,m+1}\left(y\,|\,\boldsymbol{x}\right)$$

*for every early-exit $m$ and for every input $\boldsymbol{x}$.*

*Proof.* Using the definition of the PoE ensemble and that of the Heaviside activation function, the predictive likelihood at the $m$th early exit can be written as

$$p_{\Pi,m}\left(y\,|\,\boldsymbol{x}\right) := \frac{1}{Z_m} \prod_{i=1}^{m} \left[f_i(\boldsymbol{x})_y > b\right]^{w_i}, \quad Z_m = \sum_{y' \in \{1,\dots,K\}} \prod_{i=1}^{m} \left[f_i(\boldsymbol{x})_{y'} > b\right]^{w_i}$$

where $f_i(\boldsymbol{x}) \in \mathbb{R}^K$ denotes a vector of logits at the $i$-th early-exit, $b$ represents a chosen threshold, and $w_i \in \mathbb{R}^+$ represent ensemble weights. Since we assume that class $y$ is in the support of the full NN ensemble, i.e., $p_{\Pi,M}\left(y\,|\,\boldsymbol{x}\right) > 0$, it follows that

$$\prod_{i=1}^{m} \left[f_i(\boldsymbol{x})_{\tilde{y}} > b\right]^{w_i} = \prod_{i=1}^{m+1} \left[f_i(\boldsymbol{x})_{\tilde{y}} > b\right]^{w_i} = 1, \ \ \forall m = 1, \dots, M-1\,.$$

It then remains to show that the normalization constant is non-increasing, i.e., $Z_m \geq Z_{m+1}, \forall m$. To this end observe that $\left[f_{m+1}(\boldsymbol{x})_{y'} > b\right]^{w_{m+1}} \in \{0,1\}$, $\forall y' \in \mathcal{Y}$, using which we proceed as

$$Z_m = \sum_{y' \in \{1,\dots,K\}} \prod_{i=1}^{m} \left[f_i(\boldsymbol{x})_{y'} > b\right]^{w_i} \geq$$

$$\sum_{y' \in \{1,\dots,K\}} \prod_{i=1}^{m} \left[f_i(\boldsymbol{x})_{y'} > b\right]^{w_i} \cdot \left[f_{m+1}(\boldsymbol{x})_{y'} > b\right]^{w_{m+1}} = Z_{m+1},$$

where $Z_m = Z_{m+1}$ if $\left[f_{m+1}(\boldsymbol{x})_{y'} > b\right]^{w_{m+1}} = 1 \ \forall y' \in \mathcal{Y}$ and $Z_m > Z_{m+1}$ otherwise. $\qquad\square$

# B  Additional Experiments

## B.1  Conditional Monotonicity

In the main paper, the ground-truth probability was our focus when studying conditional monotonicity in early-exit networks. Here, we supplement these findings with additional results, utilizing alternative estimators of prediction quality $\gamma_m$.

### B.1.1  Full-Model Prediction Probability

Suppose only an unlabeled hold-out dataset is available. In that case, the anytime properties of EENNs can still be investigated by relying on the full model prediction $\hat{y} := \arg\max_y p_M(y \mid \boldsymbol{x})$, instead of $y^*$. Naturally, the merit of this alternative hinges on the underlying model being accurate in most cases. Furthermore, besides serving as a substitute for the ground-truth probability in the absence of a labeled dataset, the evolution of the full-model prediction probability can be used as a stand-alone measure of how well early-exits in an EENN approximate the full model.

In Figure 7, we present conditional monotonicity results using $\hat{y}$ probability as the prediction quality measure. Similar to the results presented in Section 6.1 for $y^*$ probability, we conclude that our PA outperforms both the baseline models and CA approach across all datasets considered.

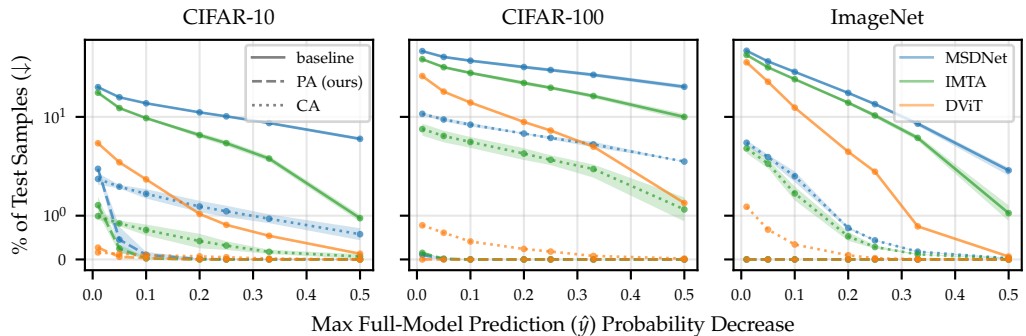

Figure 7: The percentage of test examples with a full-model prediction probability drop exceeding a specified threshold. PA significantly improves the conditional monotonicity properties of ground-truth probabilities across various datasets and backbone models. To better illustrate the different behavior of various methods, a log scale is used for y-axis. We report average monotonicity together with one standard deviation based on $n = 3$ independent runs. For DViT [Wang et al., 2021], we were unable to perform multiple independent runs and report the results for various model instantiations instead; see Appendix B.11.

### B.1.2  Correctness of Prediction

We next shift our focus to the correctness of prediction as an estimator of performance quality. Recall from Section 2 its definition: $\hat{\gamma}_m^c(\boldsymbol{x}) := [\arg\max_y p_m(y \mid \boldsymbol{x}) = y^*]$. It is a binary measure that can only take on the values of 0 or 1, and thus provides a less nuanced signal about the evolution of the performance quality compared to probability-based measures that evolve more continuously in the interval $[0, 1]$.

To aggregate the results of the conditional monotonicity analysis, we consider the following two quantities:

- $\%_{\text{mono}}$: the percentage of test data points that exhibit fully monotone trajectories in the correctness of the prediction, i.e., $\hat{\gamma}_m^c(\boldsymbol{x}) \leq \hat{\gamma}_{m+1}^c(\boldsymbol{x}), \forall m$. This implies that if the model managed to arrive at the correct prediction at the $m$th exit, it will not forget it at later exits. An oracle model with perfect conditional monotonicity will result in $\%_{\text{mono}} = 100\%$.

- $\%_{\text{zero}}$: the number of points for which the EENN model is wrong at all exits. This is used to account for trivial solutions of the monotonicity desideratum: $\hat{\gamma}_m^c(\boldsymbol{x}) = 0, \forall m$. Naturally, the smaller the $\%_{\text{zero}}$, the better.

Table 1: Aggregated results of conditional monotonicity analysis based on correctness of prediction as a performance quality measure. MSDNet [Huang et al., 2018] is used as a backbone model for PA and CA for the results reported here. See Section B.1.2 for definition of $\%_{\mathrm{mono}}$ and $\%_{\mathrm{zero}}$.

|  | CIFAR-10 | | CIFAR-100 | | ImageNet | |
|---|---|---|---|---|---|---|
|  | $\%_{\mathrm{mono}}$ ($\uparrow$) | $\%_{\mathrm{zero}}$ ($\downarrow$) | $\%_{\mathrm{mono}}$ ($\uparrow$) | $\%_{\mathrm{zero}}$ ($\downarrow$) | $\%_{\mathrm{mono}}$ ($\uparrow$) | $\%_{\mathrm{zero}}$ ($\downarrow$) |
| *MSDNet* | 91.8 | **3.6** | 70.8 | **12.6** | 84.9 | **20.4** |
| *PA (ours)* | 96.2 | 4.6 | 87.2 | 16.4 | 93.5 | 24.1 |
| *CA* | **97.8** | 5.5 | **90.8** | 18.6 | **94.5** | 25.6 |

Results are displayed in Table 1. Comparing our PA model against the MSDNet baseline, we observe a significant improvement in terms of the percentage of test data points that exhibit perfect conditional monotonicity, i.e., $\%_{\mathrm{mono}}$. However, this improvement comes at a cost, as fewer points have a correct prediction for at least one exit. Despite this trade-off, the magnitude of improvement in $\%_{\mathrm{mono}}$ is larger compared to the degradation of $\%_{\mathrm{zero}}$ across all datasets considered, which demonstrates that our PA method improves the conditional monotonicity of $\hat{\gamma}_m^c(\boldsymbol{x})$ in a non-trivial manner.

Although the Caching Anytime (CA) method outperforms PA in terms of $\%_{\mathrm{mono}}$ across all datasets, it is worth noting that a considerable portion of CA's outperformance can be attributed to trivial solutions (i.e., $\hat{\gamma}_m^c(\boldsymbol{x}) = 0$, $\forall m$), which is reflected by CA's poor performance in terms of $\%_{\mathrm{zero}}$. Furthermore, the fact that neither CA nor PA fully achieves conditional monotonicity in prediction correctness (i.e., $\%_{\mathrm{mono}} = 100$) highlights that there is still potential for further improvement in endowing EENNs with conditional monotonicity.

### B.2 Product Anytime (PA) Ablations

To better understand the role of different components in our proposed PA method, we conduct several ablations and present the results in Figure 8. In accordance with the theoretical results we presented in Section 4.2, we observe that the PoE ensemble combined with the Heaviside activation (—) yields a model that exhibits complete conditional monotonicity in the full model prediction (i.e., $\hat{y}$) probability.[9] However, this combination also leads to a model with inferior performance in terms of overall test accuracy (see left plot). As mentioned, this occurs because the model assigns an equal probability to all classes in its support. The ReLU activation resolves this issue by performing on par with Heaviside in terms of monotonicity and on par with exponential, i.e., softmax, in terms of accuracy, thus combining the best of both worlds.

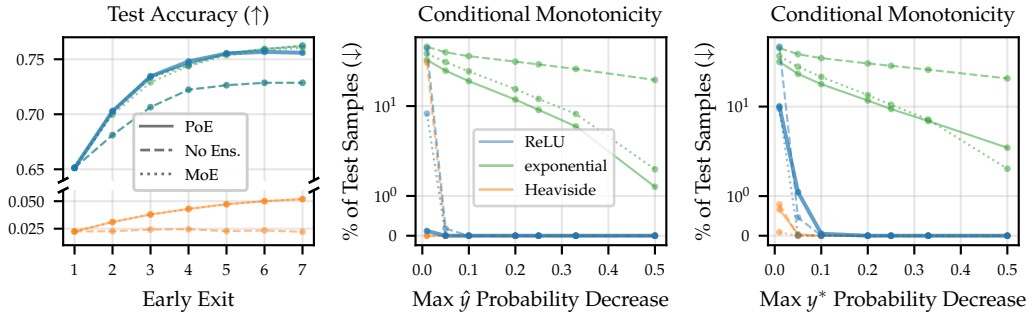

Figure 8: Test accuracy and conditional monotonicity properties of various PA model instantiations for the CIFAR-100 dataset using MSDNet as a backbone. Specifically, we vary (1) the type of activation function for mapping logits to probabilities and (2) the type of ensembling. Our proposed PA method uses ReLU activation function with product ensembling (—) and is the only method to perform well in terms of both accuracy and monotonicity.

---

[9]Note that $\hat{y} := \arg\max_y p_M(y \mid \boldsymbol{x})$ is by definition part of the support of the final classifier $p_{\Pi,M}$ (unless it degenerates to a zero distribution). However, this is not necessarily the case for $y^*$, which is the reason for a slight violation of conditional monotonicity in the PoE-Heaviside model, as seen in the right plot of Figure 8.

Delving deeper into the ReLU activation function and examining various ensembling styles, we also observe in Figure 8 that Mixture-of-Experts with ReLU (MoE-ReLU; $\cdots$) demonstrates performance comparable to our proposed PA (PoE-ReLU; —) in terms of both overall accuracy and conditional monotonicity. The MoE predictive distribution at the $m$th exit is defined as:

$$p_{\mathrm{MoE},m}\left(y \,|\, \boldsymbol{x}\right) := \sum_{i=1}^{m} w_i \cdot p_i\left(y \,|\, \boldsymbol{x}\right), \quad p_i\left(y \,|\, \boldsymbol{x}\right) = \frac{a\big(f_i(\boldsymbol{x})_y\big)}{\sum_{y' \in \{1,\ldots,K\}} a\big(f_i(\boldsymbol{x})_{y'}\big)}$$

where $a : \mathbb{R}^K \to \mathbb{R}_{\geq 0}$ represents a chosen activation function, such as ReLU, and we utilize uniform weights $w_i = 1/m$. Given this observation, one might question why we prefer PoE over MoE in our final PA model. The rationale for this decision is two-fold. First, the proof presented in Appendix A, which serves as the inspiration for our method, is only valid when PoE ensembling is employed. Second, unlike the PoE-based model, the MoE-based model does not exhibit anytime uncertainty. Instead, it demonstrates constant underconfidence across all exits, as illustrated in Figure 9. Considering these factors, we find PoE to be superior to MoE for anytime prediction purposes and incorporate it into our final proposed solution.

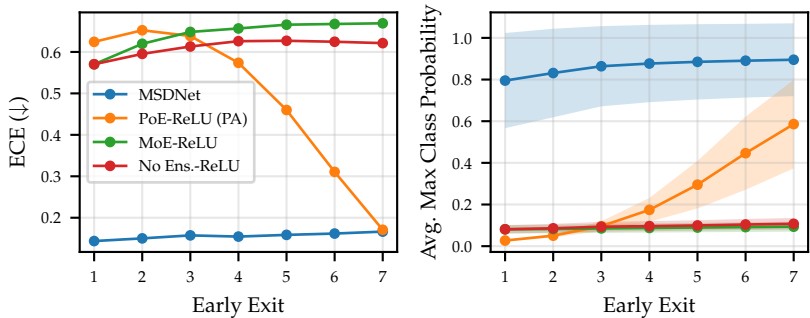

Figure 9: Expected Calibration Error (ECE; *left*) and average maximum class probability over the test dataset (*right*) for various instantiations of our PA model using the MSDNet model as a backbone on the CIFAR-100. Only product ensembling (PoE) gives rise to a model that gets progressively more confident with more early exits, as exemplified by decreasing ECE and increasing magnitude of probability. On the contrary, mixture ensembles (MoE) result in constant underconfidence.

**Early-Exit Ensembles Baselines** MoE-exponential ($\cdots$) and PoE-exponential (—) can also be interpreted as *early-exit ensemble* baselines for our PA approach. Instead of relying only on the current predictive distributions (as we do when reporting backbone EENNs results in Section 6), we ensemble distributions across all exits up to and including the current one. However, it is clear from Figure 8 that ensembling softmax predictions is insufficient for achieving conditional monotonicity, as both approaches significantly underperform compared to our PA. This underscores the importance of the activation function that maps logits to probabilities for improving monotonicity properties in EENNs. In Figure 10, we verify that these findings hold on the ImageNet dataset as well, where we again find that our PA is the only configuration that performs well both in terms of marginal accuracy and conditional monotonicity.

**Sigmoid Activation Function** So far, we have discussed why the ReLU function is preferred over the Heaviside or exponential (softmax) functions for transforming logits into probabilities in our setting. We also considered the sigmoid activation function; however, upon further examination, we identified two drawbacks that make it less suitable in our context. Firstly, the sigmoid function lacks the "nullifying" effect of ReLU. As a result, the support for classes diminishes more slowly, and we do not observe the same anytime uncertainty behavior as we do with ReLU (see Section 6.2). Secondly, logits with larger values all map to the same value, approximately 1, under the sigmoid function. This can negatively impact accuracy, as it does not preserve the ranking. Figure 11 illustrates the results using the sigmoid activation function for the CIFAR-100 dataset with MSDNet as a backbone: our choice of ReLU clearly outperforms the sigmoid activation.

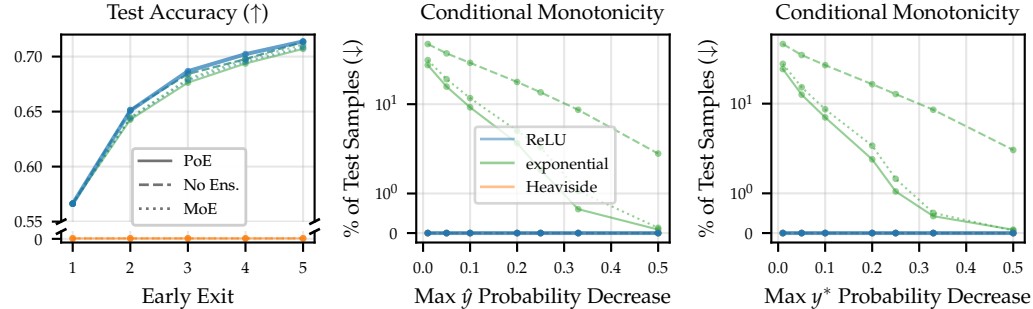

Figure 10: Test accuracy and conditional monotonicity properties of various PA model instantiations for the ImageNet dataset using MSDNet as a backbone. Our PA (—) is the only method to perform well in terms of both accuracy and monotonicity. Note that the monotonicity curves for ReLU and Heaviside activation functions (almost perfectly) overlap at $y = 0$.

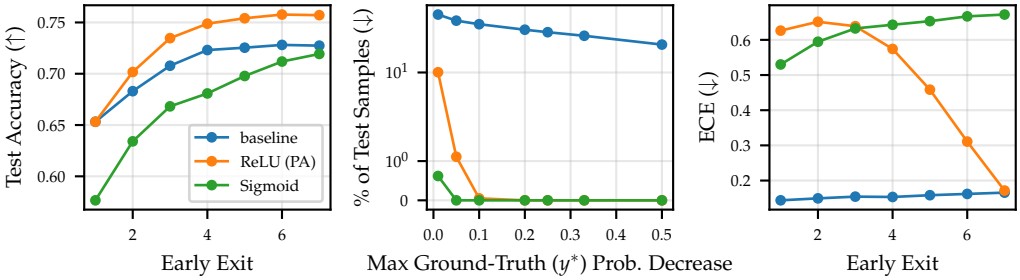

Figure 11: Test accuracy, conditional monotonicity and ECE results for MSDNet model (—) on CIFAR-100 dataset. Using sigmoid activation function (—) hurts both accuracy as well as calibration. Hence, we utilize ReLU activation in our proposed PA (—).

## B.3 Closing the Calibration Gap

Here we outline two modifications to our proposed PA method aimed at reducing the calibration gap reported in Section 6.3.

### B.3.1 Adaptive Thresholds

We first introduce an extension of our Product Anytime (PA) method, which enables navigating the monotonicity-calibration trade-off in scenarios when a hold-out validation dataset is available.

Recall that ReLU is used as an activation function in PA to map logits to probabilities. This implies that 0 is used as a threshold, determining which classes 'survive' and get propagated further, and which do not. However, 0 is arguably an arbitrary threshold, and one could consider another positive constant $b \in \mathbb{R}^+$ instead. It is important to note that the choice of $b$ directly influences the size of the support of the predictive likelihood. As illustrated in Figure 13, a larger value of $b$ results in a smaller number of classes with positive probability (as expected). This larger threshold does also lead to a reduction in the coverage of PA's predictive sets with respect to the ground-truth class. However, the

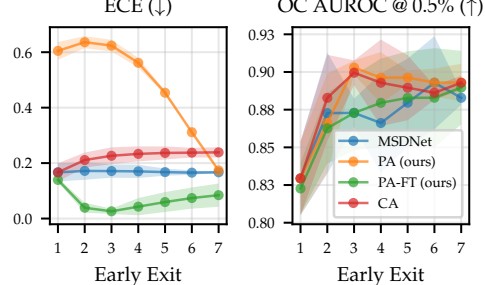

Figure 12: Calibration analysis for MSDNet on CIFAR-100. *Left*: expected calibration error (ECE) during anytime evaluation. *Right*: Oracle-Collaborative AUROC with 0.5% deferral [OC AUROC; Kivlichan et al., 2021], which simulates a realistic scenario of human-algorithm collaboration by measuring the performance of a classifier when it is able to defer some examples to an oracle, based on its maximum class probability.

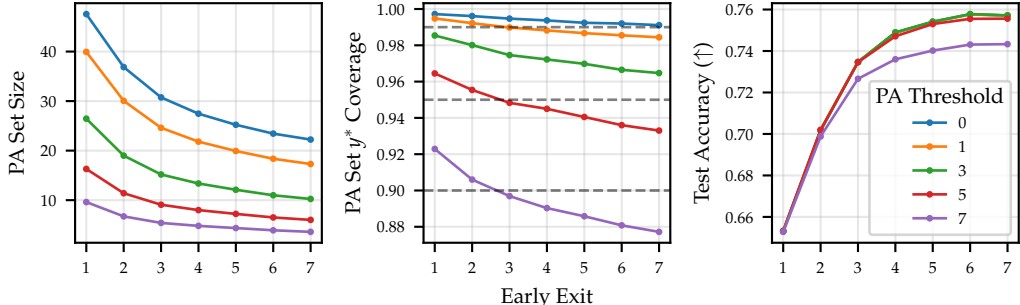

Figure 13: Static threshold analysis for PA using MSDNet as a backbone on CIFAR-100 dataset. PA's predictive set is defined as: $s_m(\boldsymbol{x}_n) := \{y \in \mathcal{Y} \mid p_{\text{PA},m}(y \mid \boldsymbol{x}_n) > 0\}$. *Right*: Average size of PA's predictive set at each exit. As expected, increasing $b$ leads to smaller sets. *Middle*: (Empirical) Coverage of ground-truth class at each exit: $c_m := \sum_n [y_n \in s_m(\boldsymbol{x}_n)] / N_{\text{test}}$. As $b$ increases, it becomes more likely for the true class $y^*$ to fall outside of the support giving rise to a smaller coverage. *Left*: The marginal test accuracy at each exit is shown as the threshold parameter $b$ varies. Increasing $b$ negatively affects accuracy, though the impact is minor for smaller values of $b$.

decrease in coverage does not significantly affect the overall accuracy of the model, as shown in the far right plot.

Expanding on this concept, one can also consider adaptive thresholding: rather than using the same threshold value for all data points, the model could dynamically adjust the threshold in the activation function based on the current example. Ideally, when the model is confident in its prediction, a higher value of $b$ is employed, resulting in smaller predictive sets. Conversely, when faced with a more challenging example, a lower threshold is utilized to capture the model's uncertainty.

Switching to ReLU with adaptive thresholding, the predictive likelihood at $i$th exit (before ensembling) is given by:

$$p_i(y \mid \boldsymbol{x}) = \frac{\max\left(f_i(\boldsymbol{x})_y, \tau\big(f_i(\boldsymbol{x})\big)\right)}{\sum_{y' \in \{1,\dots,K\}} \max\left(f_i(\boldsymbol{x})_{y'}, \tau\big(f_i(\boldsymbol{x})\big)\right)}$$

where $\tau : \mathbb{R}^K \to \mathbb{R}^+$ maps the vector of logits $f(\boldsymbol{x}) \in \mathbb{R}^K$ to a non-negative threshold.[10]

Here, we consider threshold functions of the form $\tau\big(f(\boldsymbol{x})\big) = C \cdot p_\psi\big(f(\boldsymbol{x})\big)$, where $p_\psi : \mathbb{R}^K \to [0, 1]$ is a model that estimates the probability of the datapoint $\boldsymbol{x}$ being classified correctly, and $C$ is a constant that ensures the outputs of $\tau$ have a similar magnitude to that of logits. To estimate $C$ and $\psi$, we rely on the hold-out validation dataset. Specifically, we fit a binary logistic regression model using labels $y_n^\tau := [\hat{y}_n = y_n^*]$. Additionally, we only use the $K = 3$ largest logits as input to the regression model, as we have empirically found that excluding the rest does not significantly impact the performance. To find $C$, we perform a simple grid search and select the value that yields the best monotonicity-calibration trade-off on the validation dataset. We fit a thresholding model only at the first exit and then reuse it at later exits, as this approach yielded satisfactory performance for the purposes of our analysis. It is likely that fitting a separate thresholding model at each exit would lead to further performance benefits.

In Figure 14, we observe that PA with adaptive thresholding still improves over baseline in terms of monotonicity (middle plot, note the log-scale of $y$-axis). Although the adaptive thresholding approach demonstrates worse monotonicity when compared to the PA algorithm with a static threshold (i.e., $\tau = 0$), it outperforms the PA method in terms of calibration, as measured by the Expected Calibration Error (ECE). To further elucidate the influence of adaptive thresholding, Figure 15 presents the ground-truth probability trajectories for a random selection of test data points. In general, it is apparent

---

[10]In general, $\tau$ could depend on the datapoint $\boldsymbol{x}$ itself; however, since we desire an efficient model (to avoid adding excessive overhead during inference time), we opt for a more lightweight option based on the processed input, i.e., logits.

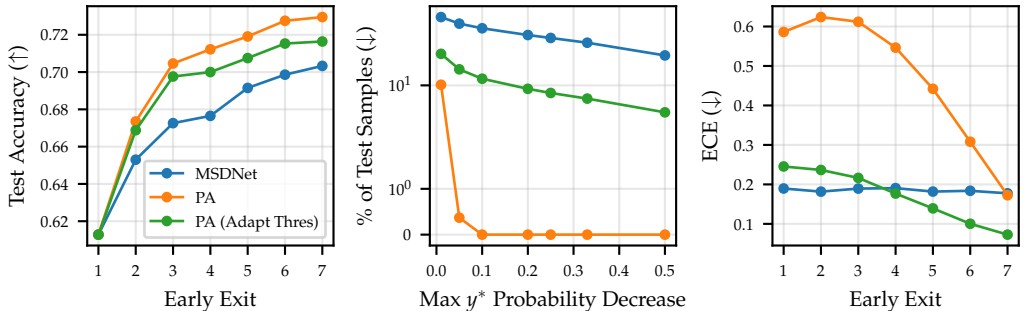

Figure 14: Results for CIFAR-100 using MSDNet as the backbone model. We see that adaptive thresholding can be effective in reducing PA's calibration gap, though at some cost to both test accuracy and conditional monotonicity. Note that the baseline results in this figure differ slightly from those in Figures 3 and 4, as less data (90%) was used to fit the model, with a portion of the training dataset (10%) used to fit the thresholding model $\tau$.

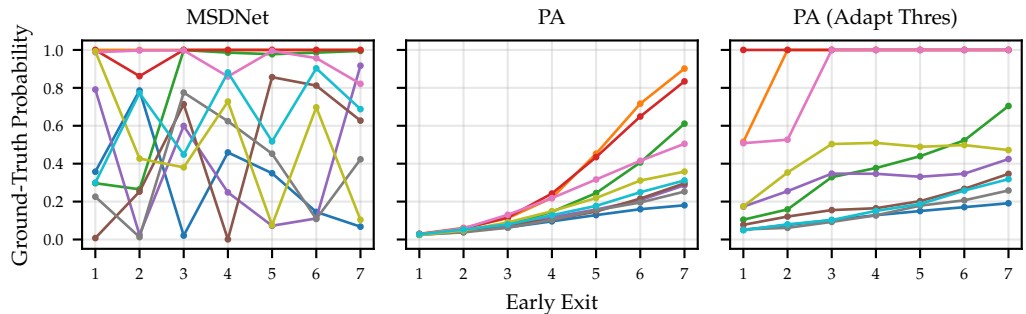

Figure 15: Ground-truth probability trajectories (i.e., $\{\hat{\gamma}_m^p(\boldsymbol{x})\}_{m=1}^M$) for 10 random test CIFAR-100 data points using MSDNet as a backbone. Both instantiations of PA yield more monotone ground-truth probability trajectories.

that adaptive thresholding can serve as an effective strategy for improving the calibration of the PA method, albeit at some expense to both conditional monotonicity and overall accuracy.

### B.3.2 PA Finetuning

The post-hoc nature of our proposed PA method is attractive due to its minimal implementation overhead. However, as mentioned in Section 6.3, it seems to adversely affect calibration, according to ECE-like metrics, when compared to baseline models. This discrepancy might be due to a mismatch between the objectives during training and testing in the PA method. To examine this hypothesis, we consider training our model using the PA predictive likelihood (as given by Equation 4) and assess its effect on calibration.

Given the product structure and the non-smooth activation function (ReLU) employed in PA's likelihood, the training can be quite challenging. To enhance the stability of the training process, we introduce two modifications. Firstly, instead of training the model from scratch, we begin by training with a standard EENN objective (see Equation 1) with a softmax likelihood and then *finetune* the model using the PA likelihood at the end.[11] Secondly, we substitute the ReLU activation function with a Softplus function[12], which we found aids in stabilizing the model's convergence during training. Consequently, we also utilize Softplus to map logits to (unnormalized) probabilities during testing. Moreover, we found that using ensemble weights $w_m = 1$ is more effective for models finetuned

---

[11]We train with EENN objective for the first $2/3$ of epochs and with PA objective for the last $1/3$ of epochs.

[12]Softplus is a smooth approximation of the ReLU function: $\text{Softplus}(x) := \log(1 + \exp x)$

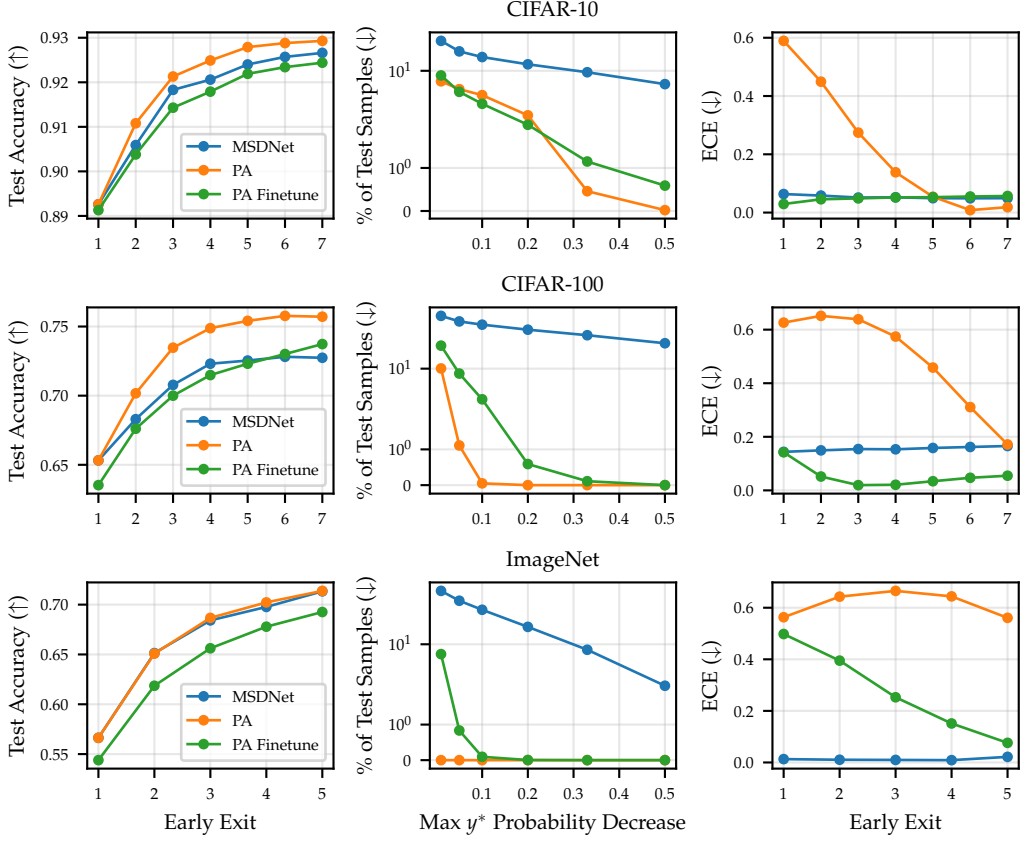

Figure 16: Results for various image classification datasets using MSDNet as the backbone model. We see that finetuning with PA (as opposed to only applying it post-hoc) can be effective in reducing PA's calibration gap, though at some cost to both test accuracy and conditional monotonicity.

with the PA objective, as opposed to our default choice of $w_m = m/M$ presented in the main paper for post-hoc PA.

The results are presented in Figure 16. Focusing initially on (marginal) test accuracy (*right*), we note that PA training (—) has a slight negative impact on performance, particularly when contrasted with the post-hoc PA (—). Similarly, it underperforms post-hoc PA in terms of conditional monotonicity (*middle*), but it considerably surpasses the baseline MSDNet model (—). When considering ECE, it becomes clear that fine-tuning with PA leads to significant improvements over post-hoc PA, thus substantiating our hypothesis that the calibration gap mainly arises from the discrepancy between train-time and test-time objectives. On the CIFAR-100 dataset, PA with finetuning even outperforms the baseline MSDNet in terms of ECE.

Based on these observations, we conclude that integrating PA during the training of EENNs is an effective strategy for bridging the calibration gap, though it incurs a minor cost in terms of overall accuracy and conditional monotonicity. We note that these costs could be mitigated (or ideally, completely eliminated) through further stabilization of product training. As such, we view this as a promising avenue for future research.

### B.3.3 Relevance of ECE in Anytime Prediction Setting

In Section 6.3, we pointed out poor Expected Calibration Error (ECE) in the initial exits as one of the limitations of our proposed PA method. It is important to note, however, that no decision in the anytime setting is based on model (or prediction) confidence, i.e., $\max_y p_m(y \mid \boldsymbol{x})$. This contrasts with the *budgeted batch classification* setting [Huang et al., 2018], where the decision on when to exit could be based on the model confidence. Thus, we believe that our PA potentially undermining

model confidence, as evident by poor ECE in the earlier exits, is not a limitation that would seriously impact the merit of our solution. Especially since we focus solely on the anytime setting, where, we would argue, non-monotone behavior in terms of performance quality is a much more serious issue.

However, it is true that model/prediction confidence, i.e. $\max_y p_m(y \mid \boldsymbol{x})$, could be useful in the anytime scenario as a proxy for model uncertainty. This allows the model—when prompted by its environment to make a prediction—to also offer an estimate of uncertainty alongside that prediction. However, given that our PA potentially adversely affects model confidence in earlier exits, we caution against its use. Instead, we recommend alternative measures of uncertainty that better align with our model, such as the conformal set size (c.f., Section 6.2).

## B.4 NLP Experiments

In this section, we conduct NLP experiments with the aim of demonstrating that our proposed approaches for adapting EENNs for anytime inference are generalizable across different data modalities.

Specifically, we employ an early-exit model from Schwartz et al. [2020], in which the BERT model is adapted for accelerated inference. The considered datasets are (1) IMDB, a binary sentiment classification dataset; (2) AG, a news topic identification dataset with $|\mathcal{Y}| = 4$; and (3) SNLI, where the objective is to predict whether a hypothesis sentence agrees with, contradicts, or is neutral in relation to a premise sentence ($|\mathcal{Y}| = 3$). We adhere to the same data preprocessing and model training procedures as outlined in Schwartz et al. [2020].

To account for the smaller number of classes in these NLP datasets, we shift logits for a (small) constant prior to applying ReLU activation function. Hence the PA likelihood at $m$th exit has the form (before applying product ensembling):

$$p_m(y \mid \boldsymbol{x}) = \max\left(f_m(\boldsymbol{x})_y + 1/C,\, 0\right)^{w_i}, \tag{5}$$

where $C$ represents the cardinality of $\mathcal{Y}$. This helps with reducing the possibility of PA collapsing to a zero distribution.

The results are depicted in Figure 17. Similar to our image classification experiments discussed in Section 6.1, we find that our proposed modifications yield comparable overall accuracy, as shown in the left plot. Considering conditional monotonicity (*middle* plot), at least one of our two suggested methods outperforms the baseline model on every dataset. However, the degree of improvement in monotonicity is considerably smaller in comparison to our image classification experiments, with the PA model even failing to yield consistent improvement on certain datasets, such as IMDB. We attribute this not to the change in data modality, but rather to the fewer number of classes in the NLP datasets used in this section as compared to the image datasets used in Section 6.1.

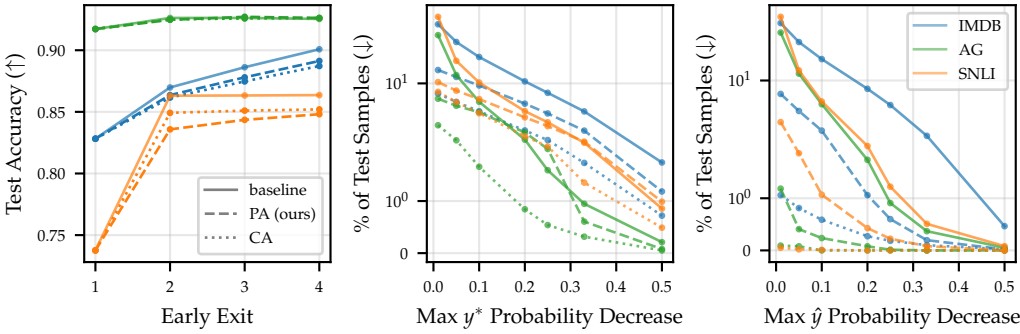

Figure 17: NLP experiments results using the BERT model from Schwartz et al. [2020] as the backbone. Despite the findings being qualitatively similar to those presented in Figures 3 and 4 for image classification datasets, the enhancement in terms of conditional monotonicity provided by our proposed methods is less pronounced for the NLP classification tasks considered in this section.

## B.5 Overthinking and Hindsight Improvability

In this section, we delve deeper into concepts that have been previously introduced in EENN literature and that share a connection with the idea of conditional monotonicity, which is the focus of our study.

In Kaya et al. [2019], the authors discovered that EENNs, for some data points, yield correct predictions early on but then produce incorrect results at later exits[13]—a phenomenon they term *overthinking*. Consequently, the authors noted that early-exiting in deep neural networks is not only beneficial for speeding up inference but can sometimes even lead to improved overall test accuracy. To measure overthinking, they considered the difference in performance between an oracle model that exits at the first exit with the correct answer[14] and the performance of the full EENN $p_M(\mathbf{y} \mid \boldsymbol{x})$. In the anytime setting, we desire a model with the lowest overthinking measure, i.e., 0, as this indicates that evaluating more layers in the early-exit network does not negatively impact performance.

In Figure 18, we present the results of the overthinking analysis for the MSDNet [Huang et al., 2018] model. We observe that both of our proposed methods significantly improve in terms of the overthinking measure (*left*), making them more applicable in anytime scenarios.

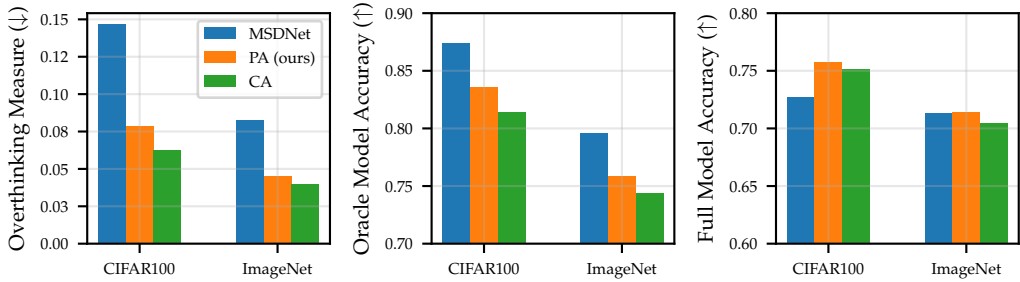

Figure 18: Overthinking analysis based on Kaya et al. [2019] using the MSDNet model. Our methods substantially improve upon the baseline MSDNet in terms of the overthinking measure (*left*), making them more suitable for an anytime setting. However, most of these improvements can be attributed to a decrease in the theoretical performance of the oracle model (*middle*) rather than to the improvement of the full EENN performance (*right*). For a more detailed explanation of the quantities depicted, please refer to Section B.5.

We also note that a decrease in the overthinking measure can result from either (1) a reduction in the (theoretical) performance of the oracle model or (2) an improvement in the (actual) performance of the full EENN. As can be seen in Figure 18 (*middle* and *right*), our proposed modifications mainly improve the overthinking measure through (1). To explore this further, in Figure 19 we plot the number of test points that the model *learned*, i.e., correctly predicted for the first time, and the number of points that the model *forgot*, i.e., incorrectly predicted for the first time, at each early exit. Both PA and CA show a clear strength in minimizing forgetting, although they do so at the cost of diminishing the baseline model's ability to learn correct predictions. [15] These results indicate that there is further room for improvement in the adaptation of EENNs for anytime computation, as the ideal model would minimize forgetting without compromising its learning abilities.

---

[13]Using notation introduced in Section 2 this implies: $\hat{\gamma}_m^c(\boldsymbol{x}) = 1$ and $\hat{\gamma}_{m'}^c(\boldsymbol{x}) = 0$ for some $m' > m$.

[14]Such a model cannot be implemented in practice, as it requires access to ground-truth values. Nevertheless, it is interesting to study it as a way to better understand the upper bound on the achievable performance of the underlying model.

[15]These observations are consistent with the analysis we presented in Section B.1.2 where we used correctness of prediction as a measure for conditional monotonicity.

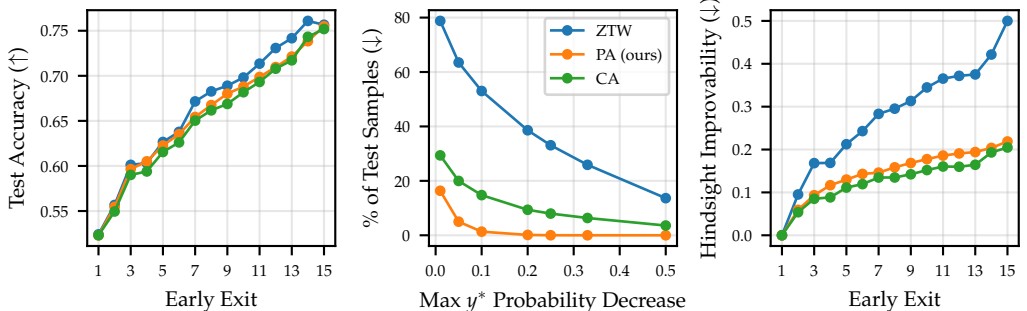

Figure 20: Results for CIFAR-100 using Zero-Time-Waste (ZTW; Wolczyk et al. [2021]) model. Applying our PA method to ZTW model leads to large improvements in terms of conditional monotonicity (*middle*) and hindsight improvability (*right*), while only causing a minor decrease in marginal accuracy (*left*).

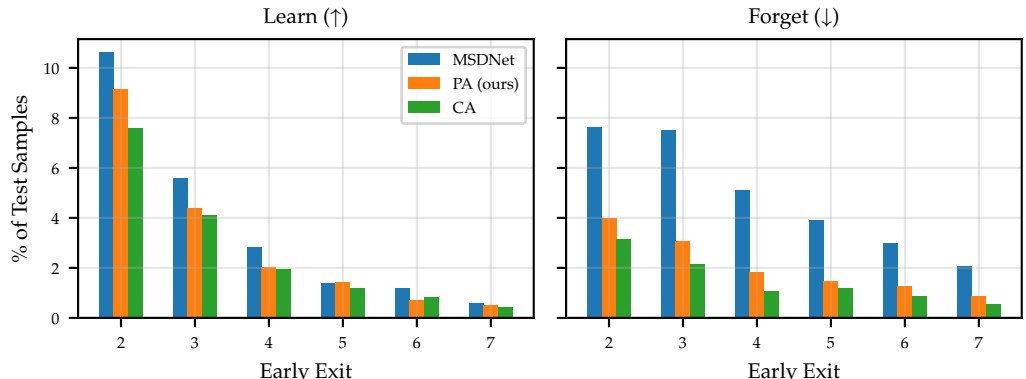

Figure 19: *Learn-vs-forget* analysis for the CIFAR-100 dataset using the MSDNet model. Specifically, we plot the number of data points that are correctly classified for the first time (*left*) and the number of data points that are incorrectly predicted for the first time (*right*) at each early exit. We observe that our modifications help the model 'forget' less, though this comes with a minor trade-off in the model's ability to 'learn'.

In a follow-up study by Wolczyk et al. [2021], the hindsight improvability (HI) measure is introduced. For each early exit, this measure takes into account the set of incorrectly predicted points and computes the percentage of those that were accurately predicted at any of the previous exits. In an anytime setting, an ideal model would consistently exhibit an HI measure of 0. As evidenced by the results in Figure 20, our post-hoc modifications substantially enhance the HI measure (*right* plot) while only causing a minor negative impact on overall accuracy (*left* plot). Similar to the overthinking measure, the observation that $HI > 0$ for our proposed methods suggests a potential for further improvements.

## B.6 Deep Ensembles and Regression

Although we have presented our results for classification with early-exit neural networks, we can easily extend them to handle generic deep ensembles and regression settings.

**Deep Ensembles**   While applying our method to deep ensembles might seem strange since they have no implicit ordering as in the case of early-exit NNs, we note that ensembles are often evaluated sequentially in low resource settings where there is not enough memory to evaluate all of the ensemble members concurrently. In such settings, there is an implicit, if arbitrary, ordering, and we may still require anytime properties so that we can stop evaluating additional ensemble members at any point. Therefore, we can apply both Product Anytime and Caching Anytime—as in Section 4—to deep ensembles, where $p_m\left(y \mid \boldsymbol{x}\right)$ now represents the $m$-th ensemble member, rather than the $m$-the early exit. An earlier version of this work [Allingham and Nalisnick, 2022] focused on product anytime in the deep ensembles case. However, in that work, the focus was also on training from scratch rather than the post-hoc setting we consider in this work. Training from scratch is challenging due to, among other things,[16] instabilities caused by the Product-of-Experts and the bounded support of the likelihoods. Thus, the PA results from Allingham and Nalisnick [2022] are worse in terms of predictive performance than the standard ensemble baseline. Note that our theoretical results in Appendix A also apply in the deep ensembles case.[17] Figure 21 shows qualitative results for our *anytime ensembles* applied to a simple classification task. As expected, the support shrinks as we add more multiplicands to the product.

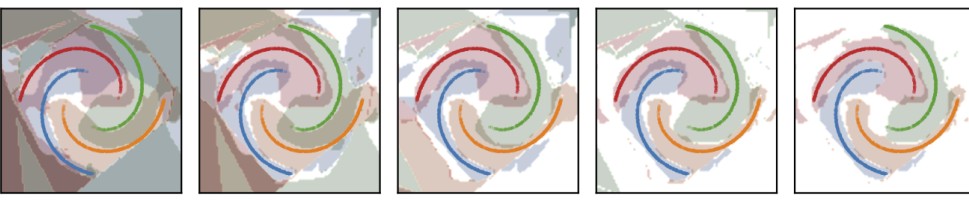

Figure 21: Evolution of the support for each of 4-classes (■, ■, ■, ■) in a spiral-arm classification task with a 5-member anytime ensemble, as 1 to 5 (left to right) members are included in the prediction.

**Regression**   Product Anytime can be applied to regression by using a uniform likelihood at each early-exit (or ensemble member)

$$p_m\left(y \mid \boldsymbol{x}\right) = \mathcal{U}(a_m,\, b_m)$$

which results in a product likelihood

$$p_{1:m}\left(y \mid \boldsymbol{x}\right) = \mathcal{U}(a_{\max},\, b_{\min})$$

where $a_{\max} = \mathtt{max}(\{a_m\})$ and $b_{\min} = \mathtt{min}(\{b_m\})$. That is, the lower and upper bounds of the product are, respectively, the maximum-lower and minimum-upper bounds of the multiplicands. As we add more multiplicands, the width of the product density can only get smaller or stay the same, and thus the probability of the modal prediction $p_{1:m}\left(\tilde{y} \mid \boldsymbol{x}\right)$ can only increase or stay the same, as in the classification case. Figure 22 shows qualitative results for anytime ensembles applied to a simple regression task. We see that the order of evaluation is unimportant. Firstly, as expected, the final result is the same, and in both cases, the support size always decreases or is unchanged. Secondly, the fit has almost converged within a few evaluations, and the last few multiplicands tend to have smaller effects. Note that the prediction bounds cover the data well even for the poorer early fits. Finally, note that there is no support outside of the training data range—we consider this data as out-of-distribution.

---

[16]For example, Allingham and Nalisnick [2022] use a mixture of the product and individual likelihoods as their loss function to encourage the individual ensemble members to fit the data well. However, they use the standard softmax likelihood—with an exponential activation function—for the individual ensemble members, which conflicts with the Heaviside activations in the product likelihood and results in reduced performance.

[17]We note that Allingham and Nalisnick [2022] neglect an important property required for monotonicity of the modal probability: $p_m\left(y \mid \boldsymbol{x}\right)$ must be uniform in addition to having finite support. I.e., in the classification setting, the activation function used to map logits to probabilities must be the Heaviside function.

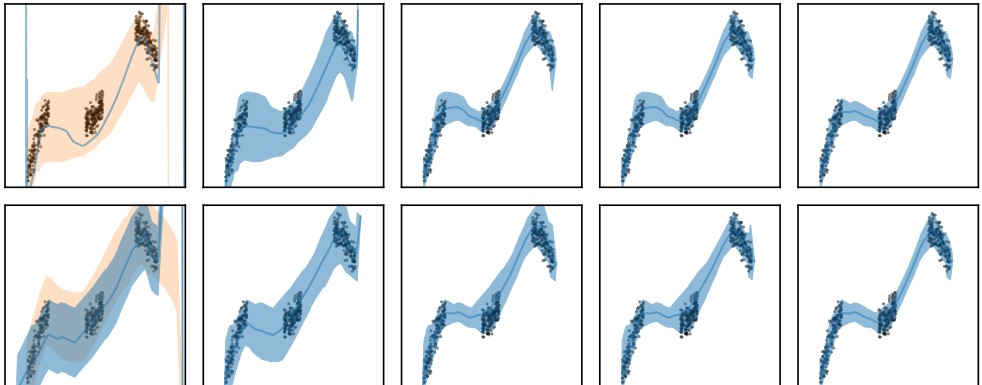

Figure 22: Evolution of the the product support (■) and mode (——) for the *simple 1d* [Antorán et al., 2020] regression task with a 5-member anytime ensemble, as 1 to 5 (left to right) members are included in the prediction. In each column the support of the multiplicand to be included in the next column's product are shown (■). The top and bottom show two different arbitrary orderings of the ensemble members.

## B.7 Anytime Uncertainty: Conditional Monotonicity

In Section 6.2, we presented the results of our uncertainty experiments where we observed that, marginally, our PA model demonstrates an *anytime uncertainty* pattern: the model starts with high uncertainty early on and then progressively becomes more confident at later exits.

In this section, we supplement these experiments with an analysis at the conditional level. Specifically, for a given $x \in \mathcal{X}$, we compute the uncertainty (for instance, entropy) at each early exit: $\{U_m(x)\}_{m=1}^M$. Following the approach we used in Section 3 for performance quality sequences, we define a *maximum increase* in uncertainty as $\text{MUI}(x) := \max_{m'>m} \{ \max(U_{m'}(x) - U_m(x), 0) \}$. Next, for a range of suitably chosen thresholds $\tau \in \mathbb{R}_{\geq 0}$, we compute the number of test instances that exhibit an uncertainty increase surpassing a given threshold $N_\tau := \sum_n [\text{MUI}(x_n) \geq \tau]$. Note that a monotone oracle model would exhibit entirely non-increasing uncertainty trajectories, meaning $N_\tau = 0, \forall \tau$.

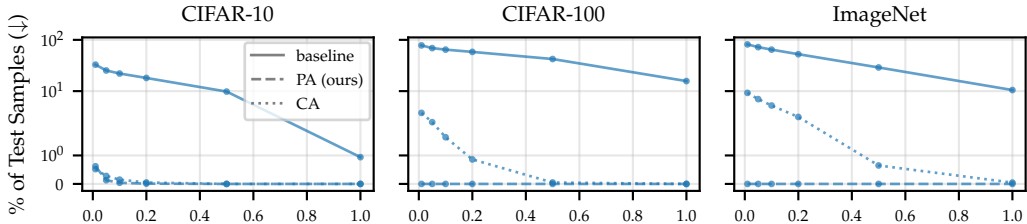

Figure 23: Conditional monotonicity using entropy as an uncertainty measure and MSDNet model as a backbone. To better illustrate the different behavior of various methods, a log scale is used for the y-axis. Our PA exhibits a significantly more monotone behavior compared to the baseline MSDNet [Huang et al., 2018] model.

The results for entropy and conformal set size are depicted in Figures 23 and 24, respectively. We observe that an EENN like MSDNet exhibits non-monotone behavior also in its uncertainty estimates across different exits. This means that there are data points where the model starts with high confidence in the initial exits but becomes less certain in subsequent stages. Such a trend is problematic in the anytime setting, similar to the issues of non-monotonicity in performance quality (c.f., Section 3). It suggests that the model expends additional computational resources only to diminish its confidence in its response. Reassuringly, our PA aids in ensuring monotonicity not only

in terms of performance quality (e.g., ground-truth probability) but also in its uncertainty estimates (e.g., conformal set size) as evident in Figures 24 and 23.

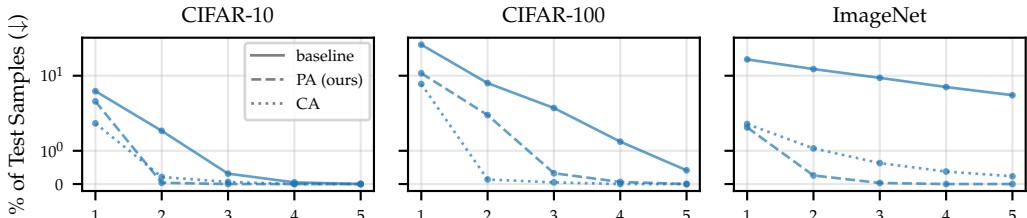

Figure 24: Conditional monotonicity using conformal set size as an uncertainty measure and MSDNet model as a backbone. To better illustrate the different behavior of various methods, a log scale is used for the y-axis. Same as in Figure 5, we used Regularized Adaptive Predictive Sets algorithm [RAPS; Angelopoulos et al., 2021] to compute conformal scores. 20% of test examples were used as a hold-out set to calculate conformal quantiles at each early exit. Our PA exhibits a significantly more monotone behavior compared to the baseline MSDNet [Huang et al., 2018] model.

## B.8    Controlling the Violations of Conditional Monotonicity

In Section 4.2, we introduced our Product Anytime (PA) method. It relies on the ReLU activation function to map logits to (unnormalized) probabilities, which unfortunately compromises the strong conditional monotonicity guarantees that are present when using the Heaviside activation function.

In this section, we propose a technique that allows for a more nuanced control of violations of conditional guarantees. To illustrate this, observe first that ReLU and Heaviside are specific cases of the following activation function:

$$a(x) := C_b \cdot \max\big(\min(x, b),\, 0\big) \tag{6}$$

where $C_b := \max(1, 1/b)$. Notice how $\lim_{b \to \infty} a(x) = \text{ReLU}(x)$ and $\lim_{b \to 0^+} a(x) = \text{Heaviside}(x)$. Therefore, by adjusting the clipping threshold $b$, we can navigate the accuracy-monotonicity trade-off as needed. This is demonstrated on the CIFAR-10 dataset in Figure 25.

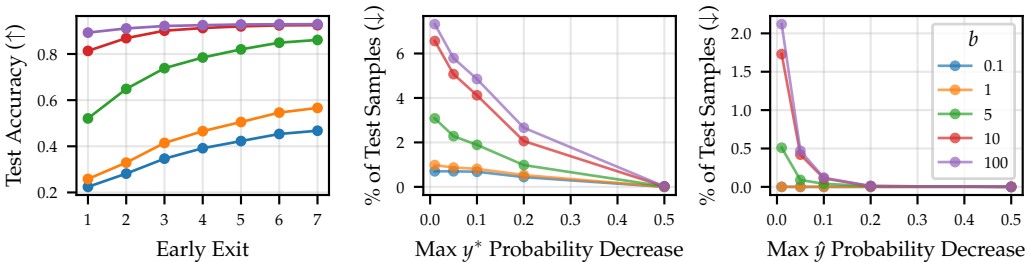

Figure 25: Test accuracy (*left*) and conditional monotonicity (*middle* and *right*) on CIFAR-10 dataset for various instantiations of the PA model with the MSDNet backbone. We vary the clipping threshold $b$ used in the activation function when mapping logits to probabilities. When $b$ is close to 0, the activation function approximates the Heaviside function, resulting in more monotone behavior, although this comes with a decrease in (marginal) accuracy. As $b$ is increased, the activation function moves closer to ReLU, leading to more violations of monotonicity, but also enhancing accuracy.

## B.9 Impact of Calibration on Conditional Monotonicity

In this section, we study whether monotonicity could be achieved by calibrating the underlying EENN rather than employing our proposed PA. To answer this, we conducted an experiment where we calibrated MSDNet and analyzed its effect on monotonicity. For the calibration, we utilized three standard techniques: temperature scaling [Guo et al., 2017], deep ensembles [Lakshminarayanan et al., 2017], and last-layer Laplace [Daxberger et al., 2021]. In Figure 26, we present results for MSDNet on CIFAR-100. While all three approaches lead to better calibration in terms of ECE (right plot), our PA significantly outperforms them in conditional monotonicity (middle plot). Moreover, all three baselines are (arguably) more complex compared to our PA: temperature scaling requires validation data, deep ensemble has M-times more parameters (M being the ensemble size), and Laplace is significantly slower compared to PA since we need test-time sampling to estimate the predictive posterior at each exit.

However, simply improving calibration might not be expected to improve monotonicity. Thus, in Figure 26, we also provide results for combining the above uncertainty calibration techniques with our CA baseline. We see that this combination provides further improvements w.r.t. monotonicity -– the improved calibration allows for better caching of the correct prediction. However, the monotonicity of these caching-and-calibrated methods still underperforms compared to PA despite being more complicated. This demonstrates that monotonicity can not be achieved via calibration alone and further underlines the value of our proposed approach.

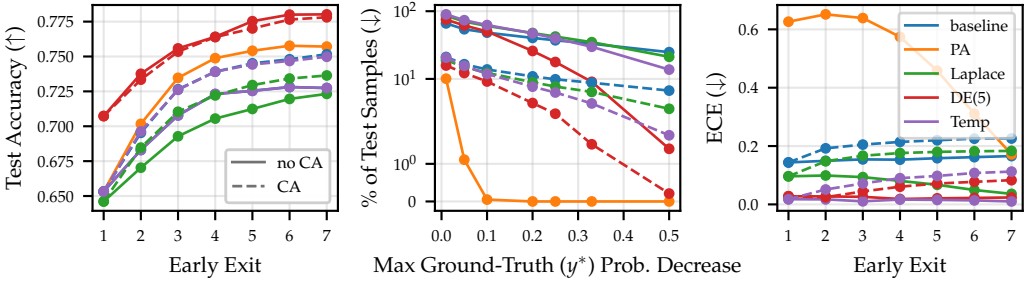

Figure 26: Comparison of our PA with calibrated baselines using MSDNet as a backbone on CIFAR-100 dataset. For calibration, we use last-layer Laplace, deep ensembles with $M = 5$, and temperature scaling. Additionally, we consider both caching (—) and non-caching (- -) versions of calibrated baselines. None of the considered calibrated baselines outperforms our PA in terms of monotonicity (middle), despite being more complex. The blue and purple solid lines overlap in the accuracy plot (*left*), indicating that the temperature scaling does not affect accuracy for the non-cached version, as expected.

## B.10 Budgeted Batch Classification

In this section, we conduct a budgeted batch classification experiment using MSDNet [Huang et al., 2018] as the backbone. In budgeted batch classification, a limited amount of compute is available to classify a batch of datapoints. The goal is to design a model that can allocate resources smartly between easy and hard examples to maximize the overall performance, such as accuracy.

The results are shown in Figure 27. Since MSDNet outperforms our PA at most computational budgets, we conclude that the monotonicity is less beneficial in budgeted batch classification. However, as seen on CIFAR datasets, the non-monotonicity of MSDNet is also concerning in this setting – for larger values of FLOPs, MSDNet performance starts to drop, while PA keeps monotonically increasing and surpasses MSDNet. Based on these results, we suggest "turning on" our PA only for anytime prediction and recommend that the user sticks to the original EENN in the budgeted batch classification. However, this certainly warrants further investigation (as seen by the performance benefits of our PA on CIFAR for larger FLOPs).

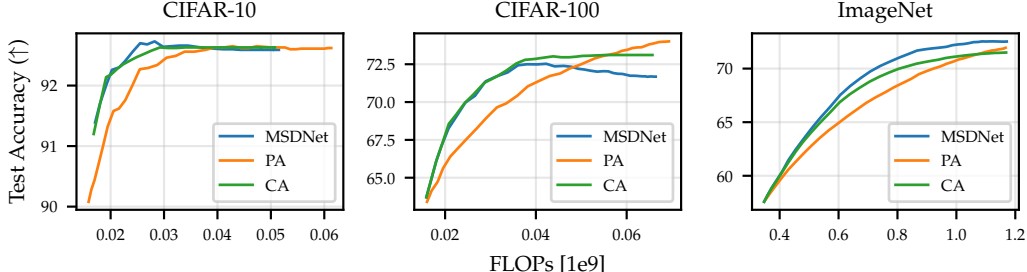

Figure 27: Budgeted batch classification results for the MSDNet model. To measure the available computational resources, we use FLOPs. Lower FLOPs will encourage the model to rely more on initial exits, while a larger number of FLOPs will encourage the model to propagate more points to the later exits. Our PA modification proves to be less beneficial in this setting, as seen by its inferior performance compared to MSDNet for most of the computational budgets.

## B.11  Additional Plots

Here, we provide additional results that were omitted in Section 6.1 to enhance the readability of the plots there. Specifically, in Figure 28, we include test accuracy results, along with their respective error bars, for the MSDNet [Huang et al., 2018] and IMTA [Li et al., 2019] models. In Figures 31 and 32, we present results for the L2W model [Han et al., 2022b]. We observe that our findings also generalize to this EENN – our PA preserves the marginal accuracy while significantly improving conditional monotonicity across all considered datasets.

As for the DViT [Wang et al., 2021] model, our attempts to train this model from scratch using the published code[18] were unsuccessful. Hence, we resorted to working with the pre-trained models provided by the authors. They only published a single pre-trained model for each model version, and as a result, instead of presenting results for independent runs with varying random seeds, we display the results for all provided DViT model instantiations in Figure 29 and 30.

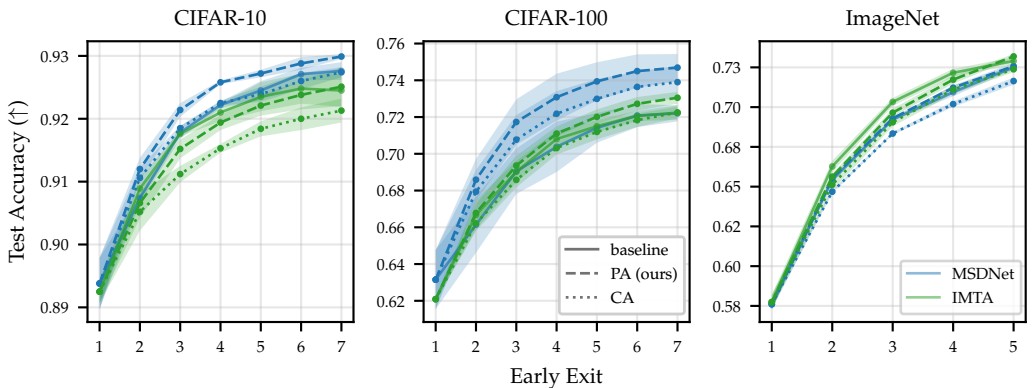

Figure 28: Test accuracy on the CIFAR and ImageNet datasets. Our PA method maintains a competitive performance compared to the baseline models. For each model and dataset, we use the same number of early exits as proposed by the authors. We plot the average accuracy from $n = 3$ independent runs.

---

[18]`https://github.com/blackfeather-wang/Dynamic-Vision-Transformer`

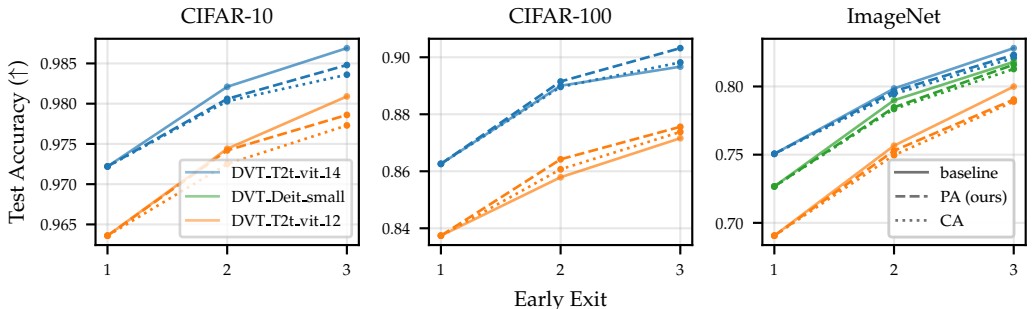

Figure 29: Test accuracy on the CIFAR and ImageNet datasets for various instantiations of DViT model [Wang et al., 2021].

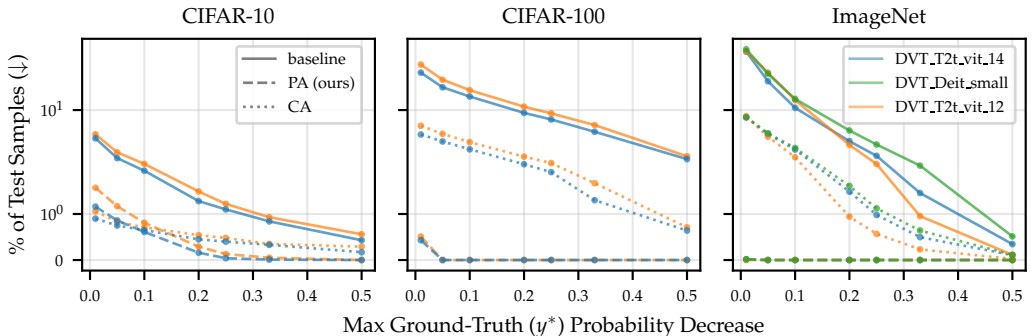

Figure 30: The % of test examples with a ground-truth probability drop exceeding a particular threshold for various instantiations of DViT model [Wang et al., 2021]. Our PA significantly improves the conditional monotonicity of ground-truth probabilities across various datasets and backbone models. To better illustrate the different behavior of various methods, a log scale is used for the y-axis.

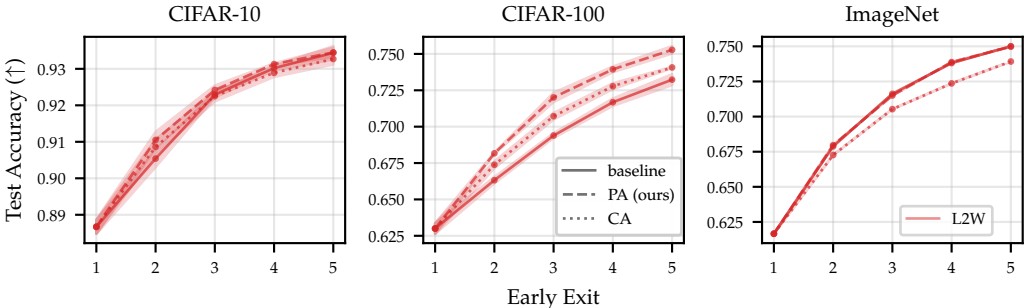

Figure 31: Test accuracy on the CIFAR and ImageNet datasets for L2W model [Han et al., 2022b]. We plot the average accuracy from $n = 3$ independent runs.

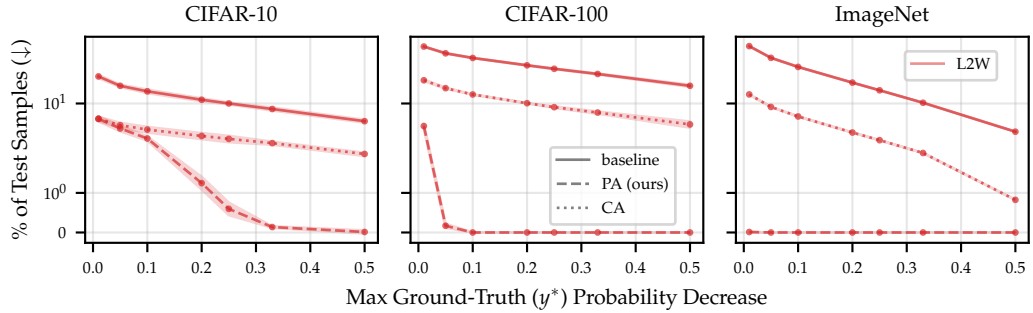

Figure 32: The % of test examples with a ground-truth probability drop exceeding a particular threshold for L2W model [Han et al., 2022b]. Results are averaged over $n = 3$ independent runs. To better illustrate the different behavior of various methods, a log scale is used for the y-axis.

