# OpenReview forum: "Towards Anytime Classification in Early-Exit Architectures by Enforcing Conditional Monotonicity"
_NeurIPS.cc/2023/Conference — NeurIPS 2023 poster_

### Official Review · Reviewer_NWJy · 2023-06-29

**Soundness:** 3 good
**Presentation:** 3 good
**Contribution:** 3 good
**Rating:** 6
**Confidence:** 5

**Summary:**

This paper studies an interesting problem of early-exit classification networks when applied to anytime prediction tasks. Specifically, the authors empirically demonstrate that in an early-exit network, the prediction confidence scores of its different exits for the same sample are not monotonically increased. Then the authors propose a method to recalculate the prediction confidence of different exits. This method is shown effective in improving the conditional monotonicity of different exits' predictions.

**Strengths:**

1. The studied problem is interesting and important;
2. The proposed method is theoretically guaranteed;
3. The experiment is comprehensive, covering different architectures and datasets.

**Weaknesses:**

I have some concerns as follows:

1. The results on CIFAR datasets (Fig. 2 left and middle) are confusing: the IMTA baselines achieved lower/similar accuracy than the MSDNet baseline in Fig. 2. However, IMTA is apparently a stronger method than the original baseline. This makes the results much less convincing to me. Moreover, I did not find the ImageNet results using the proposed PA for MSDNet and IMTA (blue and green dashed lines) in Fig 3 (right). Only DVT results (orange dashed line) are presented.

2. The experiment results in Fig. 4 indeed show that the ground-truth probability trajectories monotonically increase by using the proposed PA. However, **how can such monotonicity improve the anytime prediction task**, is not discussed. Is there any metric that can measure the anytime performance of a model? Now the authors only show us that the overall accuracy can be preserved (or improved a little).

3. It is recommended to conduct experiments on more recent multi-exit models, such as L2W-MSDNet [1].

4. I'm curious whether better monotonicity is beneficial in the budgeted dynamic inference setting (the smooth accuracy-FLOPs curves reported in MSDNet and IMTA papers).

[1] Han, Yizeng, et al. "Learning to Weight Samples for Dynamic Early-Exiting Networks." ECCV 2022.

**Questions:**

See weakness.

**Limitations:**

As I stated in the weakness part, the benefits of the obtained better monotonicity are not discussed in depth.

---

> ### Author Rebuttal · Authors · 2023-08-08
>
> We appreciate your time and efforts in reviewing our work.
>
> > the authors empirically demonstrate that in an early-exit network, prediction confidence scores [...] for the same sample are not monotonically increased.
>
> We would point out that our findings indicate the ground-truth class probability, denoted as $p_m(y^* | \boldsymbol{x})$, exhibits non-monotonic behavior. This means that there are instances where the ground-truth class is more probable in earlier exits than in subsequent ones. We believe that this is even more concerning than non-monotonicity observed in prediction confidence, represented as $\max_y p_m(y | \boldsymbol{x})$. Nevertheless, we concur that any form of non-monotonicity is troubling.
>
> For example, in Appendix B.7 we show that MSDNet is non-monotone also w.r.t. its estimates of uncertainty. Specifically, there are points where the model is highly confident in the initial exits and then becomes less confident later on. As a measure of uncertainty, we have employed entropy and conformal set sizes, but we expect the results to also hold for prediction confidence.
>
> Encouragingly, our PA helps with monotonicity not only when considering performance quality (e.g., ground-truth probability), but also for monotonicity in the uncertainty (e.g., conformal set size).
>
> > IMTA is apparently a stronger method than the original baseline. This makes the results much less convincing to me.
>
> >  It is recommended to conduct experiments [using] L2W-MSDNet
>
> We share your surprise that IMTA does not clearly outperform MSDNet in our experiments, which seems to contradict the claims in their respective papers. However, it's important to emphasize that we used the code and training scripts exactly as provided by the authors of IMTA (note that the pre-trained models were not published for this model): https://github.com/kalviny/IMTA. Our own attempts to improve the performance of the baseline IMTA model (to match the performance reported in their paper) were unsuccessful.
>
> We followed your advice and also ran the experiments using L2W model. As shown in Figure R.4 (see attached PDF), our findings remain consistent — **the PA method maintains marginal accuracy while substantially improving conditional monotonicity for L2W**. We will add this baseline to Figures 2 & 3 in the camera-ready version. We believe the inclusion of L2W will make our empirical findings stronger so we would like to thank you for this suggestion.
>
> For now we have results only for ImageNet model, because, regrettably, the authors of L2W did not release neither training code nor pre-trained models for CIFAR-10/100:  https://github.com/LeapLabTHU/L2W-DEN. We will reach out to them regarding this so that we can, hopefully, include L2W results for all considered datasets in our camera-ready.
>
> > I did not find the ImageNet [PA results] for MSDNet and IMTA [...] in Fig 3 (right).
>
> The monotonicity curves of the PA method for the MSDNet and IMTA models on ImageNet overlap with the one for the DViT model at y=0. We recognize that the current presentation might lead to confusion, and we appreciate you highlighting this issue. We plan to add a statement to the caption of Figure 3 to clarify this.
>
> >  how can such monotonicity improve the anytime prediction task, is not discussed.
>
> > the benefits of [...] better monotonicity are not discussed in depth.
>
> The primary argument of our paper is that with our PA method, we can maintain the marginal accuracy while simultaneously ensuring conditional monotonicity. This is crucial in a truly anytime setting (where the stopping time is random and determined by the environment) as it ensures that the performance will not deteriorate with more computational resources.
>
> As a concrete example of how monotonicity can help with anytime predictions, we can consider again the scenario of Android phones introduced in Section 1 (c.f. lines 21-24 and 29-32). **Deploying the original MSDNet might result in certain data points receiving inferior predictions on a higher-spec device compared to a device with limited computational capabilities**. However, if we apply our PA to an MSDNet, such inconsistencies become far less likely due to the model's monotonic behavior.
>
> >  Is there any metric that can measure the anytime performance of a model?
>
> Estimating the performance of an anytime algorithm in a real-world setting is tricky as there are many potentially important facets of "performance" to consider. We believe that the combination of marginal accuracy (Figure 2) and conditional monotonicity (Figure 3) that we focus on in our paper represent some of these facets. Unfortunately, to the best of our knowledge, previous work has almost exclusively focused on marginal accuracy, so this is an underdeveloped area. Nonetheless, we have identified two metrics proposed in the previous work that could potentially be useful for studying anytime models. The first is the "Overthinking" metric introduced in [1], which looks at difference in performance between an oracle model that exits at the first exit with the correct answer and the performance of the full model (i.e., final exit). The second, termed "Hindsight Improvability" in [2], gauges the efficiency of the early-exit network in leveraging previous predictions. We show that our PA leads to improvements over baseline MSDNet in terms of both of those metrics, see Appendix B.5 for more details.
>
> > I'm curious whether better monotonicity is beneficial in the budgeted dynamic inference setting
>
> Please refer to the *Anytime Prediction vs. Input-Dependent/Efficient Inference setting* in the global rebuttal.
>
> If you believe we have adequately addressed your concerns, we would be grateful if you would consider raising your score.
>
> [1] Kaya, Y., et al., 2019, May. Shallow-deep networks: Understanding and mitigating network overthinking. *ICML*
>
> [2] Wołczyk, M., et al., 2021. Zero time waste: Recycling predictions in early exit neural networks. *NeurIPS*

---

> > ### Comment · Reviewer_NWJy · 2023-08-15
> >
> > 1. It's strange that IMTA performs inferior to MSDNet, because I have run the experiments on CIFAR-100. Only the finetuning stage can increase the accuracy. This downgrades my confidence in your results.
> >
> > 2. Most of my concerns are addressed. About the application on L2W, I think ImageNet results are sufficient.
> >
> > 3. I'm still curious about the benefits of better monotonicity in the budgeted dynamic inference setting. Testing without finetuning or re-training is easy, taking very little time. I'd be glad to see this result.

---

> > > ### Author Response · Authors · 2023-08-15
> > >
> > > We thank you for your response.
> > >
> > > > 3. I'm still curious about the benefits of better monotonicity in the budgeted dynamic inference setting. Testing without finetuning or re-training is easy, taking very little time. I'd be glad to see this result.
> > >
> > > We have performed the budgeted batch classification experiment from MSDNet paper. Since we can not attach plots to our response here, we summarize the results below in a table format (we report test accuracy in % at various level of computational budget measured in FLOPs). We will include a full plot in the camera-ready version.
> > >
> > > **CIFAR-10**:
> > >
> > > | FLOPs[1e7] | MSDNet | PA |
> > > | --- | --- | --- |
> > > | 1.7 | **91.39** | 90.71 |
> > > | 2.0 | **92.26** | 91.58 |
> > > | 3.2 | **92.66** | 92.49 |
> > > | 4.5 | 92.59 | **92.62** |
> > > | 5.1 | 92.59 | **92.63** |
> > >
> > > **CIFAR-100**:
> > >
> > > | FLOPs[1e7] | MSDNet | PA |
> > > | --- | --- | --- |
> > > | 1.7 | **64.87** | 64.15 |
> > > | 2.0 | **68.26** | 66.4 |
> > > | 3.3 | **71.96** | 69.91 |
> > > | 4.9 | 72.20 | **72.44** |
> > > | 6.6 | 71.68 | **73.95** |
> > >
> > > **ImageNet**:
> > >
> > > | FLOPs[1e8] | MSDNet | PA |
> > > | --- | --- | --- |
> > > | 3.7 | **58.87** | 58.49 |
> > > | 5.1 | **64.44** | 62.82 |
> > > | 8.7 | **71.68** | 69.48 |
> > > | 10.5 | **72.46** | 71.23 |
> > > | 11.6 | **72.51** | 71.94 |
> > >
> > > Since MSDNet outperforms our PA at most computational budgets, we conclude that the monotonicity is less beneficial in budgeted batch classification. However, as evident on CIFAR datasets, non-monotonicity of MSDNet is worrisome also in this setting - for larger values of FLOPs, MSDNet performance starts to drop, while PA keeps monotonically increasing and surpasses MSDNet.
> > >
> > > Moreover, note that in our paper we suggest “turning on” our PA paper only for the anytime prediction - we make no claims regarding the use of our model in budgeted batch classification. The results here suggest that the user is better off sticking to the original model in the budgeted batch classification, though this certainly warrants further investigation (as evident by performance benefits of our PA on CIFAR for larger FLOPs).
> > >
> > > We will add a new section in the Appendix of our camera-ready version describing this experiment. We believe that it will be a nice addition and should further help with illustrating the difference between the two settings (anytime prediction vs budgeted batch classification). We thank you for this suggestion.
> > >
> > > > 1. It's strange that IMTA performs inferior to MSDNet, because I have run the experiments on CIFAR-100. Only the finetuning stage can increase the accuracy. This downgrades my confidence in your results.
> > >
> > > We sincerely regret the diminished confidence you have in our experimental results due to the reproducibility challenges surrounding IMTA. We are committed to addressing this by engaging with the authors of the IMTA model for clearer guidance on achieving the accuracy numbers they reported in their work.
> > >
> > > We'd like to emphasize that even if we were to replicate the IMTA results precisely (meaning that marginal accuracy would be between 0.5-2% higher compared to what we managed to get using that model), it would not have any significant impact on the overall message of our paper. Our core finding is that early-exit models are non-monotone and hence not directly applicable in anytime prediction setting. This is further supported by the fact that models outperforming IMTA in terms of marginal accuracy, such as DViT and L2W, are still highly non-monotone.

---

> > > > ### Comment · Reviewer_NWJy · 2023-08-16
> > > >
> > > > Thanks. It is interesting that the monotonicity even harms the performance in the budgeted classification setting. It is encouraged to include more (theoretical and empirical) analysis. So, to my understanding, the monotonicity is more useful in the anytime prediction setting, because it ensures that once we allow more inference time, the overall accuracy must be increased. This is the most valuable part of the proposed method. Am I right?

---

> > > > > ### Author Response · Authors · 2023-08-16
> > > > >
> > > > > > It is encouraged to include more (theoretical and empirical) analysis.
> > > > >
> > > > > We agree that a deeper analysis of the impact of monotonicity in the budgeted batch classification regime is warranted. However, this is out of scope for the current paper where we explicitly focus on the anytime setting. Nonetheless, we will make sure to thoroughly explain all the budgeted batch classification experiments we have performed based on your suggestions in a new Appendix section in camera-ready, hoping it will provide some solid pointers for future research.
> > > > >
> > > > > > So, to my understanding, the monotonicity is more useful in the anytime prediction setting, because it ensures that once we allow more inference time, the overall accuracy must be increased. This is the most valuable part of the proposed method. Am I right?
> > > > >
> > > > > Correct! And another valuable part of our proposed method is its post-hoc nature. With just 4 lines of python code (see our response to reviewer kJKM), and with no re-training, we can go from a model developed for budgeted batch classification, e.g. MSDNet, to a more monotone model suitable for the anytime regime.

---

> > > > > > ### Comment · Reviewer_NWJy · 2023-08-17
> > > > > >
> > > > > > Thanks, I have raised my rating. I look forward to seeing the revised version including the additional results and discussion.

---

> > > > > > > ### Author Response · Authors · 2023-08-17
> > > > > > >
> > > > > > > Thanks for raising your score! We are grateful for your engagement in this discussion, it surely helped us improve our work.

---

### Official Review · Reviewer_ne1V · 2023-07-04

**Soundness:** 3 good
**Presentation:** 3 good
**Contribution:** 2 fair
**Rating:** 5
**Confidence:** 4

**Summary:**

This submission focuses on multi-exit early networks, from the perspective of anytime inference that can accommodate varying computational budgets. Such models produce a progressive refinement of the final prediction, offering the opportunity to "exit-early", providing a meaningful prediction. The main motivation of this work is to enforce conditional monotonicity on the predictions of successive early-exits, such as the output quality consistently improves as the inference process progresses. The proposed solution adopts the Product-of-Experts formulation and can be applied post-training, to achieve monotonicity on prediction quality as well as prediction uncertainty.

**Strengths:**

-The submission identifies and studies an interesting challenge with early-exit models, in the context of anytime inference, which is often overlooked by relevant literature.

-The papers analysis is wide and insightful, providing numerous empirical results that can motivate further research in the topic.

-The proposed methodology is clearly explained, and experiments are conducted on a range of traditional benchmarks, consistently to the practise of relevant literature.

**Weaknesses:**

-Certain design choices are not adequately justified and further comparisons/ablations should be included on the main paper. More specifically, it will be beneficial to examine: i) how the proposed methodology would behave in frozen backbone early-exit models and ii) in comparison to simple ensemble between early-exit models that comprises a intuitive baseline.

-Additionally, it is claimed that the proposed approach can also be applied at training time, but this scenario is not adequately evaluated. Would the proposed methodology incur additional error or under-confidence on the  shallower exits (instead of improving the deeper ones) in a quest to achieve the desired monotonicity? And if so how can this be mitigated?

-Figure 2-right may suggest a scalability issue for the proposed approach as the baseline dominates the depth-accuracy trade-off in contrast to other datasets. This limitation should be further investigated.

-The motivation of progressively increasing predictive confidence, may contradict the general benefits of uncertainty estimation in early-exit models (e.g. for input dependent inference). One could argue that uncertainty should ideally be calibrated with the probability of the ground-truth class; rather than penalised on early-classifier even if the provided prediction is correct.

-Finally, the discussion of the results (e.g. on Fig. 2) could benefit from providing some quantitative conclusions on text (e.g. average/median improvement etc.).

**Questions:**

In accordance to the discussion in "Weaknesses" please clarify:
1. How does the proposed methodology compare to vanilla early-exit ensembles in terms of both accuracy and uncertainty monotonicity (should be added in Fig.2 analysis).
2. Is the proposed methodology still relevant if applied on early-exit models trained via the frozen-backbone methodology (i.e. is the behaviour of Fig1-middle still evident in this case; and resolved by the proposed approach)?
3. Is there a scalability issue when the proposed method is applied on ImageNet? What aspect (sample size, number of classes etc) seem to mostly cause this effect ?
4. Is the predictive accuracy and uncertainty of early-classifier reduced when applying the proposed methodology at training time ?
5. Is it indeed beneficial to decrease prediction confidence on shallow classifiers, even when their predictions are correct (e.g. on easier samples) ?
6. In the case of the IMTA early-exit model, where knowledge distillation is applied, it is expected that the prediction uncertainty of the resulting model will be affected. How does this affect the analysis of Sec.6.2; do the results remain consistent for IMTA models ?

**Limitations:**

Some limitations are clearly discussed in a dedicated subsection of the paper. Following the above discussion, it may be beneficial to broaden this discussion accordingly.

---

> ### Author Rebuttal · Authors · 2023-08-07
>
> We would like to thank you for your valuable review and many interesting questions.
>
> > how the proposed methodology would behave in frozen backbone early-exit models
>
> > Is the proposed methodology still relevant if applied on early-exit models trained via the frozen-backbone methodology
>
> **Yes, our proposed PA is still relevant when applied to EENNs with frozen-backbone**. For a concrete example, see Appendix B.5, specifically Figure 16, where we present the results using ZTW [1]. ZTW begins with a pretrained backbone, e.g., ResNet, and adds early-exit heads on top of it. During training of the exit heads, the backbone is frozen (see Algorithm 1 in their paper). Figure 16 shows that ZTW is highly non-monotone, corroborating our findings in Section 3. Furthermore, applying our PA to ZTW substantially outperforms the original ZTW in terms of Hindsight Improvability, a metric proposed by the ZTW authors [1] to capture how effectively a model reuses past predictions.
>
> If by "frozen backbone early-exit models" you're referring to something else, please clarify, and we'll discuss further. If there's a specific EENN you're interested in, let us know and we'll do our best to conduct the relevant experiments during the discussion period.
>
> > in comparison to simple ensemble between early-exit models that comprises a intuitive baseline.
>
> > How does the proposed methodology compare to vanilla early-exit ensembles
>
> We did explore this baseline, though we didn't label it as “early-exit ensembles”. We'll clarify this in the camera-ready. The results are in Appendix B.2, see the MoE-exponential (corresponding to an ensemble of all softmax predictives up-to-and-including the current exit) in Figure 7. **In short, vanilla early-exit ensembles are not sufficient to achieve monotonicity**.
>
> For now, the results are only available for the CIFAR-100 with MSDNet. If you think it would be beneficial for us to extend these experiments to other datasets and models, we would be happy to do so and include the results in Appendix B.2. We did not to include the early-exit ensembles baseline in Figures 2 & 3, since these plots are already somewhat busy. If you disagree with this choice, we are willing to change it.
>
> > certain design choices are not adequately justified and further comparisons/ablations should be included
>
> Please let us know if there are any specific additional ablations we should perform. As discussed with reviewer G2Sm, we’ll include a more comprehensive ablation on the activation function as well as on the ensemble weighting scheme in the camera-ready.
>
> > Is the predictive accuracy and uncertainty of early-classifier reduced when applying the proposed method at training time?
>
> Due to space constraints, this didn't make it into the main text. However, we intend to incorporate it using the extra page allocated for the camera-ready. Please see Appendix B.3.2 , where we elaborate in detail on the application of our PA model during training/finetuning. As shown in Figure 12, fine-tuning with PA does slightly compromise both accuracy and monotonicity compared to the post-hoc PA. Nonetheless, it still markedly surpasses the baseline model in terms of monotonicity and largely closes the calibration gap of the post-hoc PA. We perceive this as a promising avenue for future exploration.
>
> > Is there a scalability issue when […] applied on ImageNet?
>
> We are not entirely sure what you mean by scalability issue here, let us know if we’ve misinterpreted.
>
> If by scalability issue you mean that our method leads to a decrease marginal accuracy for DViT and IMTA models on ImageNet, we would point out that the performance drop is rather marginal (amounting to less than 1%). Considering that a lack of monotonicity can be detrimental in the anytime setting, we deem such a decline in marginal performance as an acceptable trade-off for a significantly more monotone model.
>
> > Is it indeed beneficial to decrease prediction confidence on shallow classifiers, even when their predictions are correct (e.g. on easier samples) ?
>
> > progressively increasing predictive confidence, may contradict the general benefits of uncertainty estimation in EENNs (e.g. for input dependent inference).
>
> We agree that decreasing confidence also on easier samples could be problematic in input-dependent inference. However, this application is out of scope of our current paper. Please refer to *Anytime Prediction vs. Input-Dependent/Efficient Inference setting* in the global rebuttal for further discussion.
>
> We also encourage you to have a look at Appendix B.3.1, where we introduce PA with adaptive thresholding. As shown in Figure 11, this adaptation does not reduce confidence for all data points early on, but rather for the more challenging ones. However, monotonicity is compromised when compared to vanilla PA; see Figure 10. Moreover, it isn’t post-hoc due to the fitting of a thresholding model. For these reasons, the main text prioritizes the original PA. Nonetheless, we believe that adaptive thresholding presents a promising avenue for future work.
>
> > […] uncertainty should ideally be calibrated with the probability of the ground-truth class; rather than penalised […] even if [...] prediction is correct.
>
> We agree with this concern and have discussed it in Section 6.3 of the paper. Please also refer to *Calibration Gap* in the global rebuttal.
>
> >  How does this affect the analysis of Sec.6.2; […] consistent for IMTA models ?
>
> We followed your suggestion and have replicated experiment from Section 6.2 using IMTA model. As depicted in Figure R.3 (see attached pdf), the uncertainty results are consistent for this baseline model.
>
> > the discussion  […] could benefit from some quantitative conclusions [...]
>
> Will do!
>
> If you think we have sufficiently answered your questions, we would appreciate if you would consider increasing your score.
>
> [1] Wołczyk, M., et al., 2021. Zero time waste: Recycling predictions in early exit neural networks. *NeurIPS*

---

> > ### Comment · Reviewer_ne1V · 2023-08-15
> >
> > Thank you for providing thorough and constructive clarifications to all raised comments. I acknowledge I have read them, along with the comments of the other reviewers.
> >
> > I believe it would be beneficial to incorporate some corresponding changes to the manuscript, to ensure that the above assumptions, limitations and insights are clearly stated.
> >
> > Additionally, including a more comprehensive comparison with MoE on ImageNet (both in terms of accuracy and monotonicity) would significantly increase the conclusiveness of this comparison.

---

> > > ### Author Response · Authors · 2023-08-16
> > >
> > > Thank you for your response and help in improving our work.
> > >
> > > We will, of course, incorporate all the valuable feedback from the reviews into our camera-ready version as we firmly believe this will further strengthen our manuscript.
> > >
> > > We will add a comparison with MoE baseline for all the models and datasets to Appendix B.2. Since we unfortunately can not attach plots to our response here, we summarise the results for MSDNet model on ImageNet in a table format below
> > >
> > > **Test accuracy [%] per early exit**
> > >
> > > |  | 1 | 2 | 3 | 4 | 5 |
> > > | --- | --- | --- | --- | --- | --- |
> > > | MSDNet | 58.0 | 65.2 | 69.3 | 71.3 | 72.1 |
> > > | MSDNet-PA | 58.0 | 65.6 | 69.3 | 71.7 | 72.7 |
> > > | MSDNet-MoE | 58.0 | 64.7 | 68.7 | 70.9 | 72.3 |
> > >
> > > **Conditional Monotonicity [%] per decrease threshold**
> > >
> > > (the lower the better)
> > >
> > > |  | 0.05 | 0.1 | 0.2 | 0.33 | 0.5 |
> > > | --- | --- | --- | --- | --- | --- |
> > > | MSDNet | 46.8 | 34.0 | 18.5 | 8.1 | 2.3 |
> > > | MSDNet-PA | 0.0 | 0.0 | 0.0 | 0.0 | 0.0 |
> > > | MSDNet-MoE | 17.1 | 8.6 | 2.5 | 1.1 | 0.1 |
> > >
> > > As seen here, this results confirm our findings from Figure 7 that vanilla early-exit ensembles are not sufficient to make the model fully monotone (contrary to our PA).

---

> > > > ### Comment · Reviewer_ne1V · 2023-08-16
> > > >
> > > > Thank you for providing these additional results so timely. In my opinion, these indeed validate the previous findings with respect to the MoE vs PA comparison.
> > > >
> > > > As such, I am planning to update my review, following the reviewer discussion, accordingly.

---

> > > > > ### Author Response · Authors · 2023-08-17
> > > > >
> > > > > Thank you for your willingness to update your review. We're also grateful for the insightful questions you posed during the review process. Integrating your feedback will help us further enhance the quality of our paper.

---

> > > > > ### Comment · Area_Chair_AWU4 · 2023-08-21
> > > > >
> > > > > Dear reviewer,
> > > > >
> > > > > The discussion period has ended. If you wish to update your review as you mentioned in the comment, please do so by Aug 22 EOD.
> > > > >
> > > > > Thank you

---

### Official Review · Reviewer_kJKM · 2023-07-06

**Soundness:** 2 fair
**Presentation:** 3 good
**Contribution:** 2 fair
**Rating:** 4
**Confidence:** 4

**Summary:**

The authors propose a new approach, called Product Anytime (PA), for monotonic confidence estimation in early-exit neural networks. Inspired by the Product-of-Experts (PoE) approach, PA takes the normalized confidence with respect to the product of ReLU thresholded prediction logits. The authors show through experiments that PA achieves similar accuracy curves as the original softmax logits, but improves the monotonicity of early-exited prediction confidences.

**Strengths:**

1. The authors tackle a practical and important problem that should be solved in early-exit neural networks.
2. The intuition of monotonicity from the Product-of-Experts (PoE) makes sense to me.
3. The provided evaluations are comprehensive and support the claims made in the paper.

**Weaknesses:**

1. Although I agree with the authors monotonicity is an important property in early-exit NNs. I am not completely convinced whether it is meaningful to design a specialized algorithm to achieve it. My point is, can we use the confidence/uncertainty calibration algorithms [1, 2] in the literature, that seem to produce relatively accurate confidence estimation to replace the proposed algorithm? On one hand, the uncertainty calibration algorithm better reflects the accuracy of the model prediction. At the same time both approaches (uncertainty estimation and the proposed algorithm) can not provide theoretical guarantee as PoE does. I think uncertainty quantification should provide better monotonicity than softmax confidence. On the other hand, uncertainty calibration gives better intuition on which sample should be early exited, since it does not have the "calibration gap" issue observed in the proposed PA method.
2. I am unsure how the proposed algorithm should be used in the practical runtime deployment scenario.
3. The proposed approach seems to have multiple obvious limitations, such as the zero distribution due to non-overlapping labels among stages, the poor confidence at initial stages, and suboptimal performance for applications with the limited number of classes.


[1] Gal, Yarin, and Zoubin Ghahramani. "Dropout as a Bayesian approximation: Representing model uncertainty in deep learning." international conference on machine learning. PMLR, 2016.

[2] Lakshminarayanan, Balaji, Alexander Pritzel, and Charles Blundell. "Simple and scalable predictive uncertainty estimation using deep ensembles." Advances in neural information processing systems 30 (2017).

**Questions:**

What is the unique advantage of the proposed approach compared to conventional uncertainty estimation algorithms applied in early-exit neural networks?

**Limitations:**

Yes, the authors have a separate section to discuss the limitations of the proposed technical solution.

---

> ### Author Rebuttal · Authors · 2023-08-08
>
> We thank you for your time in reviewing our work. We address your concerns below.
>
> > The authors propose [PA] for monotonic confidence estimation
>
> We would just point out that our model primarily targets monotonicity in the ground-truth class probability, denoted as $p_m(y^* | \boldsymbol{x})$, and not in prediction confidence, represented as $\max_y p_m(y | \boldsymbol{x})$. Nevertheless, we concur that any form of non-monotonicity is troubling. Encouragingly, our PA helps with monotonicity not only when considering performance quality (e.g., ground-truth probability), but also for monotonicity in the uncertainty (e.g., conformal set size, see also Appendix B.7 for a more detailed analysis on this).
>
> > Although I agree with the authors monotonicity is an important property in early-exit NNs. I am not completely convinced whether it is meaningful to design a specialized algorithm to achieve it.
>
> > I am unsure how the proposed algorithm should be used in the practical runtime deployment scenario.
>
> Respectfully, we would disagree that our solution is particularly specialized. The method we propose is entirely post-hoc, meaning it can be readily applied to any pre-trained early-exit neural network. The implementation consists of four lines of Python code:
>
> ```python
> # logits: torch tensor with shape (t, C) where t represents index of current exit and C number of classes
> # weight: PoE weights (torch tensor with shape (t,) )
>
> probs = torch.clamp(logits, min=0) # ReLU activation
> probs = probs.pow(weights[:, None])  # apply PoE weights
> probs = torch.prod(probs, dim=0)  # product ensemble
> probs /= probs.sum()  # normalize
> ```
>
> **Thus, any model originally designed for an efficient inference scenario (e.g., MSDNet) can be repurposed for anytime prediction by simply "turning on" our post-hoc solution.**
>
> We hope that with our answer here, we have also addressed your question on the “practical runtime deployment scenario”. If not, please let us know, and we will be happy to provide additional details and clarifications.
>
> > On the other hand, uncertainty calibration gives better intuition on which sample should be early exited
>
> We would point out that in the anytime setting the “intuition on which sample should be early exited ” is of lesser importance since the environment dictates the exit point, not the user. Please refer to *Anytime Prediction vs. Input-Dependent/Efficient Inference setting* section in the global rebuttal for more.
>
> > My point is, can we use the confidence/uncertainty calibration algorithms [1, 2] in the literature, that seem to produce relatively accurate confidence estimation to replace the proposed algorithm?
>
> > What is the unique advantage of the proposed approach compared to conventional uncertainty estimation algorithms applied in early-exit neural networks?
>
> Please refer to *Monotonicity via Calibration* section in the global rebuttal above, where we have performed a new experiment based on your questions. If you feel that our answers there inadequately address your concerns, please let us know, and we will be happy to elaborate further.
>
> > The proposed approach seems to have multiple obvious limitations, such as the zero distribution due to non-overlapping labels among stages, the poor confidence at initial stages, and suboptimal performance for applications with the limited number of classes.
>
> Please refer to the *Calibration Gap section* in the global rebuttal for a longer discussion on “the poor confidence at initial stages”.
>
> Regarding the issue with “zero distribution due to non-overlapping labels among stages” - while we agree that this could be concern, we have not observed many issues with this in our experiments. **The collapse to a zero-distribution was found to occur extremely rarely on the considered datasets**. Specifically, when using MSDNet, we observe this scenario for 3 (out of 10000) test cases for CIFAR-10, and never for CIFAR-100 and ImageNet. Also, in such cases, we provide a concrete suggestion of falling back to softmax probabilities, which we found to work well in practice.
>
> To counter the issue of suboptimal performance when dealing with a small number of classes (i.e., < 5), we recommend using the Caching Anytime (CA) approach instead of the PA method. Although CA was originally proposed in Zilberstein’s work [1], it has often been overlooked in contemporary anytime literature, where the standard practice has been to return the most recent prediction. We hope our work will reignite interest in the CA approach within the anytime literature. Moreover, poor performance for a small number of classes is common among approaches relying on Product-of-Experts ensemble. In [2] , the authors report the same issue; see their Appendix C.4.
>
>
>
> If you think we have addressed you concerns sufficiently, we would appreciate you considering raising your score.
>
> [1] Zilberstein, S., 1996. Using anytime algorithms in intelligent systems. *AI magazine*
>
> [2] Wołczyk, M., et al., 2021. Zero time waste: Recycling predictions in early exit neural networks. *NeurIPS*

---

> > ### Comment · Reviewer_kJKM · 2023-08-15
> > **Thanks for the repsonse.**
> >
> > Some of my concerns have been addressed. However, I can not agree with the authors that in an anytime setting, the exit point should be solely decided by the environment. Instead, the "input dependent" setting is more reasonable, where the exit point is decided by both the resource constraint/environmental factor, and also the data sample itself, such that the coordination can happen between samples. Specifically, samples that achieve high ground-truth class probability can be early exited to save time for low ground-truth class samples, while their overall throughput or average inference speed still satisfies the resource constraint. This is a more intelligent strategy to improve the general prediction quality than "letting the environment dictate the exit point".

---

> > > ### Author Response · Authors · 2023-08-15
> > >
> > > We thank you for your reply.
> > >
> > > In our work, we assume that the model sees one data point at a time when deployed (in line with previous literature on anytime algorithms, see [1] and [2]). We acknowledge that we have not made this explicit enough in the current version of the manuscript. We will add some clarifying sentences to our manuscript and would like to thank you for pointing this out.
> > >
> > > In addition, we would highlight that in the seminal MSDNet paper [1], the authors have distinguished two potential uses of early-exit neural networks (EENNs).
> > >
> > > - The first, termed the “anytime prediction,” is when the environment dictates when computation should be stopped. This is equivalent to saying that the computational budget is unknown beforehand. Moreover, here the model sees a single datapoint at a time. The goal is to build a model that can use additional compute to increase the quality of the prediction, i.e., it should start with a crude initial prediction and then refine it given more resources/time.
> > > - The second, termed “budgeted batch classification” (but often also “dynamic/efficient inference” or “input dependent inference”) is when the computational budget is known beforehand. The user can hence allocate a budget for performing computation on a batch of examples, and the goal is to distribute the computational budget across the examples to maximize the overall accuracy. If our reading of your response is correct, you are referring to this setting.
> > >
> > > While we agree with the authors of MSDNet on the two proposed use cases of EENNs, we believe that important differences between the two settings, in terms of their desiderata and potential applications, have been overlooked. To address this, our work focuses on the first setting. We demonstrate that an early-exit neural network like MSDNet falls short in fulfilling a fundamental prerequisite of anytime prediction: monotonicity. Furthermore, we put forth a post-hoc solution to make EENNs more suitable in the anytime regime.
> > >
> > > Of course, *hybrid schemes* in which a batch of inputs is processed with an environment-decided overall budget are also possible. However, this is outside of the scope of our paper. Even if you are particularly interested in such a hybrid scheme, we hope you can appreciate the need for better distinction between the two settings (anytime prediction vs. budgeted batch classification) and see the value in a well-focused paper that solves one problem well.
> > >
> > > [1] Huang, G., et al., 2017. Multi-scale dense networks for resource efficient image classification. *ICLR*
> > >
> > > [2] Zilberstein, S., 1996. Using anytime algorithms in intelligent systems. *AI magazine*

---

> > > > ### Author Response · Authors · 2023-08-18
> > > >
> > > > Dear reviewer kJKM,
> > > >
> > > > we thank you again for your efforts during the reviewing process.  Please let us know if we've sufficiently addressed your concerns regarding the distinction between anytime prediction and budgeted batch classification.
> > > >
> > > > For further clarity, we'd like to share two quotes from previous research that effectively explain the different settings. Note how in both instances, the authors suggest that in an anytime prediction, the model processes a single datapoint at a time.
> > > >
> > > > In the MSDNet paper [1], the authors write in their abstract:
> > > >
> > > > *Two such settings are: 1. anytime classification, where the network’s prediction **for a test example** is progressively updated, facilitating the output of a prediction at any time; and 2. budgeted batch classification, where a fixed amount of computation is available to classify a set of examples that can be spent unevenly across “easier” and “harder” inputs.*
> > > >
> > > > Furthermore, in the introduction section of [2] authors explain
> > > >
> > > > *An anytime predictor (Horvitz 1987; Boddy and Dean 1989; Zilberstein 1996; Grubb and Bagnell 2012; Huang et al. 2018) can automatically trade off between computation and accuracy. **For each test sample**, an anytime predictor produces a fast and crude initial prediction and continues to refine it as budget allows, so that at any test-time budget, the anytime predictor has a valid result for the sample, and the more budget is spent, the better the prediction. Anytime predictors are different from cascaded predictors (Viola and Jones 2001; Xu et al. 2014; Cai, Saberian, and Vasconcelos 2015; Bolukbasi et al. 2017; Guan et al. 2017) for budgeted prediction, which aim to minimize average test time computational cost without sacrificing average accuracy: a different task (with relation to anytime prediction). Cascades achieve this by early exiting on easy samples to save computation for difficult ones, but cascades cannot incrementally improve individual samples after an exit.*
> > > >
> > > > We look forward to your response.
> > > >
> > > > [1]  Huang, G., et al., 2017. Multi-scale dense networks for resource efficient image classification. ICLR
> > > >
> > > > [2] Hu, H., et al., 2019, Learning Anytime Predictions in Neural Networks via Adaptive Loss Balancing. AAAI

---

> > > > > ### Comment · Reviewer_kJKM · 2023-08-19
> > > > >
> > > > > Thanks for your extra responses. Upon reading all of them, I prefer to keep my original rating.

---

> > > > > > ### Author Response · Authors · 2023-08-19
> > > > > >
> > > > > > Thanks for your response.
> > > > > >
> > > > > > We understand your decision to keep your original score. If your time permits, we would be interested to know which concern that you brought up made you decide so? This is not fully clear to us at the moment, and getting a clarity on this would help us a lot in improving our work. Specifically, we are interested to hear your thoughts on our interpretation that your previous response (*"However, I can not agree with the authors [...] letting the environment dictate the exit point"*) reflects more the budgeted batch classification setting rather than the anytime one (which is the focus of our work).
> > > > > >
> > > > > > Thanks in advance!

---

### Official Review · Reviewer_G2Sm · 2023-07-19

**Soundness:** 2 fair
**Presentation:** 4 excellent
**Contribution:** 2 fair
**Rating:** 6
**Confidence:** 3

**Summary:**

The paper first reviews the definition & key properties of anytime models (Zilberstein, 1996) and argues to focus on equipping the EENNs with conditional monotonicity (in addition to their built-in interruptibility, Sec 2). The paper then verifies that existing EENNs lack this critical property (Fig 1), proposes the PoE-based post-hoc modification with ReLU relaxation to encourage conditional monotonicity (Sec 4.2), and validates its accuracy (Fig 2), monotonicity (Fig 3) and uncertainty (Fig 4) on CIFAR-10/100 and ImageNet with MSDNet, IMTA and DViT. The paper also presents results on ablations (Sec B.2), calibration gap (adaptive thresholding & fine-tuning, Sec B.3), NLP (Sec B.4), ensembling, regression (Sec B.6) and different degrees of ReLU relaxation (Sec B.8).

**Strengths:**

+ [Originality] The paper novelly modifies the PoE formulation to achieve conditional monotonicity, making it sufficiently and advantageously different from existing works e.g. (Wolczyk et al., 2021).
+ [Clarity] The paper is well written and easy to follow. All figures are well made and the appendix is very informative.
+ [Quality] The paper is of good quality, with the proposed method relatively well motivated and designed, and the empirical evaluation (including results in the appendix) relatively comprehensive, although further improvements can be made (see Weaknesses).
+ [Significance] Given the importance of the topic, even though the results are not necessarily perfect (but decent enough), I think this work is reasonably promising and could aid/inspire future research.

**Weaknesses:**

- [Evaluation]
1) Given the criticality of the relaxation/approximation of the Heaviside function, it’s highly desirable to see more thorough analyses of and comparisons between different approximations, e.g. sigmoid (seemingly much more logical than ReLU), clipped ReLU, learnable Padé approximation [1], etc., in addition to Sec B.8.
2) The exponent $i/M$ of the proposed PA method (Eq 4) is clearly a powerful inductive bias ensuring the network starts with low confidence, which should deserve more analyses such as: (a) How do the baselines perform when equipped with this simple feature or corresponding adjustments on the softmax temperature? (b) How does $i/M$ compare to other progressions or constant exponents? (c) Does $i/M$ adversely limit the confidence on simpler test samples such that e.g. early exits become ineffective?

[1] Padé Activation Units: End-to-end Learning of Flexible Activation Functions in Deep Networks, ICLR, 2020.

**Questions:**

- Are $w_i$ and $b_i$ in Eq 4 learnable during fine-tuning? Why or why not?

**Limitations:**

The authors have reasonably addressed the paper’s limitations in Sec 6.3.

---

> ### Author Rebuttal · Authors · 2023-08-08
>
> We thank you for your feedback and help in improving our work.
>
> > Given the criticality of the relaxation/approximation of the Heaviside function, it’s highly desirable to see more thorough analyses of and comparisons between different approximations, e.g. sigmoid …
>
> We fully agree with your observation on the importance of the activation function. We would politely draw your attention to Appendix B.2 where we compare our choice of ReLU against Heaviside and exponential (which corresponds to softmax), as well as to Appendix B.8 where we report the results for clipped ReLU (even though we do not call it such in the current version, we will update this in the next version of our manuscript).
>
> Your suggestion regarding the sigmoid function is appreciated. Initially, we also considered it a viable option; however, upon further examination, we found it to have two drawbacks that make it less applicable in our context. Firstly, the sigmoid function lacks the "nullifying" effect of ReLU. Consequently, the support of classes diminishes more slowly, and we do not observe the same anytime uncertainty behaviour as we do with ReLU (c.f. Section 6.2). Secondly, logits with larger values all map to the same value, approximately 1, under the sigmoid function. This can negatively impact accuracy, as it does not preserve the ranking. Figure R.2 (see attached PDF) illustrates the results using the sigmoid activation function for the CIFAR-100 dataset with MSDNet as a backbone: **our choice of ReLU clearly outperforms sigmoid activation**. We will prepare an Appendix section in the camera-ready version and include these results there. Thank you for your suggestion.
>
> We want to underscore at this point that our use of activation functions deviates from their traditional usage within the deep learning literature. In the context of our work, when we refer to activation functions, we are discussing the function that maps logits to probabilities in the final layer of the network. Conventionally, however, an activation function refers to the non-linearity that dictates which neurons are propagated between layers. This distinction may not have been sufficiently clear in the current version of our manuscript. We appreciate your observation and will correct this in the camera-ready version.
>
> Regarding PAU activation [1]: thanks for pointing out this work; we were unaware of it before. However, based on our understanding, this activation has trainable parameters that are usually fitted with the rest of the NNs’ parameters. As such, it is not entirely compatible with our post-hoc approach. Related to the previous paragraph, it also seems PAU is used more as a non-linearity between layers rather than a function that maps logits to probabilities in the final layer. That being said, it's plausible that PAU might be a suitable choice for the activation function in our fine-tuning approach (see Appendix B.3.2). We plan to investigate this possibility in our future work.
>
> > 2a) How do the baselines perform when equipped with this simple feature or corresponding adjustments on the softmax temperature?
>
> We tried finding the optimal temperature for the baseline model (MSDNet), see *Monotonicity via Calibration* section in the global rebuttal above. In short, temperature scaling helps with the calibration of the baseline model. However, it still significantly underperforms our PA when it comes to achieving monotonicity.
>
> > 2b) How does $i/M$  compare to other progressions or constant exponents?
>
> > Are $w_i$ and $b_i$ in Eq 4 learnable during fine-tuning? Why or why not?
>
> We experimented with various coefficient schemes, such as:
>
> - $w_i = 1$
> - $w_i =$ np.linspace(1, L/2)[i]
> - learn $w_i$ during fine-tuning, similar to ZTW paper [2]
>
> **However, we found that none of these alternative weights consistently outperformed our choice of $w_i=i/M$** across different datasets and backbone models, both in terms of marginal accuracy and conditional monotonicity. In the camera-ready version of our paper, we will include a separate section in the Appendix detailing this ablation study on the selection of $w_i$.
>
> > 2c) Does $i/M$  adversely limit the confidence on simpler test samples such that e.g. early exits become ineffective?
>
> In the anytime setting upon which we're focusing, our method (utilizing $i/M$ coefficients) maintains the marginal accuracy (see Figure 2). As such, we conclude that our method does not render the initial exits ineffective. If your comment is directed towards the efficient/input-dependent inference setting, kindly refer to the *Anytime Prediction vs. Input-dependent/Efficient Inference Setting* section in the global rebuttal above.
>
> [1] Molina, A., et al., 2020. Páde Activation Units: End-to-end Learning of Flexible Activation Functions in Deep Networks. *ICLR*
>
> [2] Wołczyk, M., et al., 2021. Zero time waste: Recycling predictions in early exit neural networks. *NeurIPS*

---

> > ### Comment · Reviewer_G2Sm · 2023-08-16
> > **Re: Rebuttal**
> >
> > Thank you for the detailed response and additional results!
> >
> > In addition to the Android example in the paper, I think another possibly more attractive scenario would be multitasking in a (medical, industrial, etc.) system with strict real-time requirements, where the anytime model and other processes (possibly with higher priorities) compete for and get varying amount of resources (e.g. CPU cycles) per time slot (e.g. one video frame), so the anytime model must work with the environment to deliver as-good-as-possible results with the given (varying) budget.

---

> > > ### Author Response · Authors · 2023-08-17
> > >
> > > Thank you for your response!
> > >
> > > Your suggestion is much appreciated; we will incorporate it into our introduction, to further motivate the need for anytime models.

---

### Official Review · Reviewer_6rYn · 2023-07-25

**Soundness:** 2 fair
**Presentation:** 2 fair
**Contribution:** 2 fair
**Rating:** 4
**Confidence:** 3

**Summary:**

This paper proposes a post-hoc modification of early-exit neural networks. Specifically, the authors focus on the conditional monotonicity out of four properties presented in a prior work. A training-based method is also proposed to improve the model confidence.

**Strengths:**

The proposed method based on product-of-experts improves conditional monotonicity and its effectiveness is also supported by theoretical results.

**Weaknesses:**

- Some writing is somewhat verbose/redundant. For example, while properties of anytime models take a considerable amount of space, they are not that meaningful at last.

- Plots on "% of Test Samples vs. Max y* Prob. Decrease" should be described in a better way, in that what it means and why do we care about it.

- As addressed in Fig. 5, the proposed method results in hurting model confidence. It is good to see that fine-tuning with the proposed learning objective can improve this, but it makes the point of this work distracted; recall that this paper means to propose a post-hoc method.

- How PA Finetune is done is not explained enough.

- I am not sure if we really need to care about monotonicity. Enforcing to have monotonicity might be harmful, as shown in some curves in Fig. 2. Instead, assuming that the model is calibrated well, we could rely on the confidence of intermediate result and exit early regardless of the time budget for the best performance.

**Questions:**

Please address concerns in weaknesses.

--- post rebuttal

I appreciate the authors' active responses during the author-reviewer discussion period. Most of my concerns have been solved, however, the main concern on the necessity of (conditional) monotonicity for EENNs is still not yet solved. Hence, I increase my rating, but not by much.

In my opinion, when proposing a new metric, the metric itself should either 1) be meaningful by itself or 2) have a strong positive correlation with the main goal. The monotonicity is not meaningful by itself (a model producing a constant output has the perfect monotonicity, which is not useful in practice) and the correlation with the overall performance (accuracy) is not so significant. To provide successful cases as examples, ECE in confidence calibration is crucial when considering the reliability of model outputs, and forgetting in continual learning has a positive correlation with the average accuracy.

Regarding the scalability, the monotonicity measure seems scalable, but ECE is not for the proposed method. To me, confidence calibration is one of the most important topic in deep learning for a successful deployment to the real world applications, so non-scalable ECE is my major concern. In short, increasing the monotonicity leads to slightly improve the overall accuracy, while significantly deteriorating confidence calibration in the large-scale experiment on ImageNet.

However, given that conditional monotonicity is important (and given that ECE is somewhat problematic), I agree that the post-hoc method proposed by the authors is simple yet effective in terms of the proposed monotonicity metric.

**Limitations:**

Nothing special.

---

> ### Author Rebuttal · Authors · 2023-08-07
>
> Thank you for your feedback and help in improving our paper.
>
> > Some writing is somewhat verbose/redundant. For example, while properties of anytime models take a considerable amount of space, they are not that meaningful at last.
>
> While we focus exclusively on the monotonicity property, we have included a discussion on other relevant anytime desiderata (i.e., consistency and diminishing returns) to outline promising directions for future research. One of our goals in this work was to highlight that the field has largely neglected these desiderata. If you disagree with this inclusion, we are prepared to relocate the information to the Appendix.
>
> Please let us know if you identified any other instances of redundant writing. We would be happy to address these issues.
>
> > Plots on "% of Test Samples vs. Max y* Prob. Decrease" should be described in a better way, in that what it means and why do we care about it.
>
> Concerning the question, "why do we care about it," this plot is designed to capture the percentage of test datapoints that display non-monotone trajectories in performance quality. A non-monotone performance trajectory signifies that the model's performance deteriorates with more evaluated exits, a behavior that stands in contradiction to the monotonicity desideratum outlined in Section 2 (see Properties of Anytime Models). Thus, this plot enables us to compare different models with respect to their monotonicity. Specifically, models with lower monotonicity curves (i.e., closer to y=0) are more effective at utilizing additional computational budget to refine their predictions.
>
> Also, we would politely point out two paragraphs on lines 150-166 and the caption of Figure 1, where we tried to explain our conditional monotonicity plots in detail. Should you have concrete suggestions on enhancing the description of this plot further, we will gladly integrate those into the camera-ready version.
>
> > As addressed in Fig. 5, the proposed method results in hurting model confidence. It is good to see that fine-tuning with the proposed learning objective can improve this, but it makes the point of this work distracted; recall that this paper means to propose a post-hoc method.
>
> Please refer to the *Calibration Gap* section in the global rebuttal above. We must also respectfully disagree with the statement that the fine-tuning "distracts from the point of our work." We view the post-hoc nature of PA as a highly desirable property, allowing our method to be easily compatible with most existing early-exit NNs. Thus, we have emphasized the post-hoc version of PA in the main paper. However, recognizing that preserving calibration can be important in some applications, we have also proposed a fine-tuning version that helps close the post-hoc approach's calibration gap.
>
> > How PA Finetune is done is not explained enough.
>
> Due to space constraints, this didn't make it into the main text. However, we intend to incorporate it using the extra page allocated for the camera-ready. Moreover, we would like to draw your attention to Appendix B.3.2, where the PA finetuning is explained in detail. If you have specific suggestions for enhancing our description of the fine-tuning approach, we would be more than happy to incorporate them.
>
> > I am not sure if we really need to care about monotonicity
>
> We must respectfully disagree on this point. Monotonicity, as a key property of anytime models, was introduced by Zilberstein in 1996 [1]. This seminal work, which has garnered over 1000 citations since its publication, has motivated most of the modern anytime models [2, 3]. **All other reviewers recognized the value of monotonicity.**
>
> > Enforcing to have monotonicity might be harmful, as shown in some curves in Fig. 2
>
> While our PA method may slightly decrease marginal accuracy in some instances (CIFAR-10 DViT, ImageNet DViT, ImageNet IMTA), we emphasize that the performance drop is rather marginal (amounting to less than 1%), and that this is the exception not the rule. Considering that a lack of monotonicity can be detrimental in the anytime setting (since more computation is not guaranteed to improve performance), such minor decline in marginal performance (i.e., < 1%) is in our opinion an acceptable trade-off for a significantly more monotone model.
>
> > Instead, assuming that the model is calibrated well, we could rely on the confidence of intermediate result and exit early regardless of the time budget for the best performance.
>
> Please refer to the *Monotonicity via Calibration* section in the global rebuttal above. Moreover, your comment regarding the strategy of relying "on the confidence of intermediate result and exiting early regardless of the time budget for optimal performance" may be related to our Caching Anytime (CA) baseline (c.f. Section 4.1). In CA, the most confident prediction up to that point is cached and returned when prompted by the environment (instead of the latest prediction). While this approach is indeed a viable alternative to our PA method for enhancing monotonicity, our experimental results show that it falls short of PA in terms of both marginal accuracy (Figure 2) and conditional monotonicity (Figure 3) in most cases.
>
> > **Rating:** 3: Reject: For instance, a paper with technical flaws, weak evaluation, inadequate reproducibility
>
> Based on your score of 3, it appears that you perceive a technical flaw, weak evaluation, or inadequate reproducibility in our work. **We would sincerely appreciate further elaboration on these concerns, as we are confident that any such concerns can be addressed in this rebuttal period**.
>
> [1] Zilberstein, S., 1996. Using anytime algorithms in intelligent systems. *AI magazine*
>
> [2] Grubb, A., et al., 2012. Speedboost: Anytime prediction with uniform near-optimality. *AISTATS*
>
> [3] Huang, G., et al., 2017. Multi-scale dense networks for resource efficient image classification. *ICLR*

---

> > ### Comment · Reviewer_6rYn · 2023-08-15
> > **Thanks for the response**
> >
> > Thank you for your response. Below I leave my comments/questions after reading the rebuttal.
> >
> > > while properties of anytime models take a considerable amount of space, they are not that meaningful at last.
> >
> > I don't have a strong opinion on this. If authors and other reviewers agree that these desiderata other than monotonicity are worth to discuss, then the current state might be okay.
> >
> > > Plots on "% of Test Samples vs. Max y* Prob. Decrease"
> >
> > Providing an example scenario with specified numbers would be helpful to understand what it means (e.g., an EENN test case with fluctuating or monotonic y* over layers). It would also be nice if there is a metric that quantifies conditional monotonicity by a single number, similar to ECE for confidence calibration.
> >
> > > the proposed method results in hurting model confidence. (fine-tuning) makes the point of this work distracted.
> >
> > Regarding the answer of the authors "the post-hoc nature of PA as a highly desirable property," could you clarify if you want to improve the model confidence or performance over layers? To my understanding, conditional monotonicity is about the robustness of the model prediction (or confidence calibration) over layers (whether the model exhibits better confidence as going deeper) rather than improving the performance over layers (whether the model exhibits better performance as going deeper). However, the experimental results show that the proposed method hurts the calibration of the model confidence, i.e., increasing ECE.
> >
> > By the way, plots on "% of Test Samples vs. Max y* Prob. Decrease" seem to be somewhat problematic, because a model producing fixed outputs regardless of the layer index exhibits perfect conditional monotonicity.
> >
> > By looking at the experimental results in Figure 12, "PA finetune" overall decreases the test accuracy, so I wonder if PA is a well-defined learning objective that results in better optimization. In worst case, it would work as a random learning objective that bothers to learn the main task, such that it makes the model just less confidence and less accurate. Then, the improved conditional monotonicity or ECE might come from the random noise on the learning objective. To address this, authors may want to check the gradient and optimal state of the PA objective, and training a model from scratch rather than fine-tuning pre-trained one.
> >
> > > Enforcing to have monotonicity might be harmful.
> >
> > Authors responded that "Considering that a lack of monotonicity can be detrimental in the anytime setting (since more computation is not guaranteed to improve performance), such minor decline in marginal performance (i.e., < 1%) is in our opinion an acceptable trade-off for a significantly more monotone model." The proposed method does not only decreases the performance (which might be okay if the gap is not significant), but also harms confidence calibration, measured by ECE. It seems authors wanted to claim that the proposed method exhibits better uncertainty by the conformal set size comparison. In my opinion, authors had to focus more on discussion on the discrepancy between ECE and the conformal set size comparison. At this point, ECE is the only reliable metric to me.
> >
> > > (additional question 1)
> >
> > According to Figure 12, PA finetune makes the ECE strictly worse on ImageNet, i.e., the observation in Figure 5 is not scalable. I wonder how the authors think about the scalability of the proposed method, especially considering the result on Figure 12. I think early-exit scenario is meaningful mostly when we meet a large-scale problem, but it seems the observation in this work is not well-scaled to large-scale settings.
> >
> > > (additional question 2)
> >
> > In Figure R.1, why does temperature scaling affect the test accuracy? As I understand, it scales all logits by the same factor, so the final results should not be changed.

---

> > > ### Author Response · Authors · 2023-08-16
> > > **Response Part 1**
> > >
> > > Thank you for your response and further engagement. We are encouraged to note that most of your concerns from the initial review appear to have been addressed.
> > >
> > > > Providing an example scenario with specified numbers would be helpful to understand what it means […]
> > >
> > > We tried to do so in lines 153-156. If you feel this is not sufficient, or that it would be more interpretable to present ground-truth probability trajectories in a table format, let us know and we will incorporate this.
> > >
> > > > It would also be nice if there is a metric that quantifies conditional monotonicity by a single number […]
> > >
> > > We appreciate this suggestion. One suitable metric here would be area under the conditional monotonicity curve (AUC), where the lower value denotes a better, i.e. more monotone, model. Below we report this metric for CIFAR-100 dataset:
> > >
> > > |  | MSDNet | IMTA | DViT |
> > > | --- | --- | --- | --- |
> > > | baseline | 19.45 | 12.37 | 5.53 |
> > > | PA | 0.26 | 0.42 | 0.01 |
> > >
> > > We plan to incorporate AUC values in the camera-ready. However, we'll present these alongside our existing plots. We feel that the visual representation of conditional monotonicity provided by these plots offers a more intuitive understanding than a standalone metric (similar to how a reliability diagram helps understanding ECE).
> > >
> > > > plots […] seem to be somewhat problematic, because a model producing fixed outputs regardless of the layer index exhibits perfect conditional monotonicity.
> > >
> > > We fully agree with your observation here, i.e. a model that has ground-truth probability equal to 0 at each exit achieves perfect conditional monotonicity. That is precisely the reason why we never study conditional monotonicity in isolation, but always in combination with marginal accuracy. This is analogous to ECE - a random MNIST classifier that predicts each class with probability of 0.1 will achieve perfect ECE (i.e., 0). Hence, one should always study ECE in combination with accuracy.
> > >
> > > > could you clarify if you want to improve the model confidence or performance over layers? […]
> > >
> > > In our response here we assume that by model confidence you are referring to prediction confidence, i.e. $\max_y p_m(y | \boldsymbol{x})$. Let us know if we have misunderstood.
> > >
> > > Our primary aim is to improve performance quality over exits and not the model confidence (c.f. lines 118-132). What could potentially be a source of confusion here is the measure with which we estimate the performance quality, i.e. the probability of the ground-truth class $p_m(y^*|\boldsymbol{x})$. While this quantity is related to model (or prediction) confidence, the two are not equivalent. Here we would also like to draw your attention to Appendix B.1, where we show that our PA solution leads to more monotone behaviour even when other measures of performance quality are used, e.g. correctness of prediction $[\arg\max_y p_m(y|\boldsymbol{x}) = y^*]$ .
> > >
> > > However, it is true that in Section 6.2 and Appendix B.7 we additionally argue for monotonicity in the uncertainty estimates in anytime models. Since model/prediction confidence is one possible way to estimate the uncertainty,  we recognise that this could be another potential source of confusion. We will ensure to make the distinction between performance quality and confidence more explicit in the camera-ready.
> > >
> > > > […] "PA finetune" overall decreases the test accuracy, so I wonder if PA is a well-defined learning objective  […] random learning objective [...] improved conditional monotonicity or ECE might come from the random noise on the learning objective. […]
> > >
> > > We would respectfully push back here. PA-finetune achieves on-par accuracy to MSDNet on CIFAR datasets and 2-3% worse accuracy on ImageNet. While we agree that these results could be better, we find it hard to believe that this could be an indication of “random learning objective”.
> > >
> > > PA-finetune is well motivated - the calibration gap highlighted in our main paper primarily stems from the difference between the training and testing objectives. Specifically, MSDNet is trained using the softmax parametrization of the categorical likelihood, while our PA employs the ReLU version. PA-finetune directly addresses this mismatch by exposing the model to ReLU likelihood already during training. Therefore, we disagree with the notion that the enhanced ECE results from "random noise." Additionally, we provide theoretical motivation for the improved conditional monotonicity associated with our PA approaches, as detailed in Appendix A. Given this, we maintain that it's highly improbable for the improvements in monotonicity to arise merely from "random noise”.
> > >
> > > Furthermore, we attempted to train models from scratch using the PA-finetune objective. Although such models do converge—serving as additional empirical evidence against the notion of a "random learning objective"—we discovered that training from scratch doesn't outperform the finetuning method. Consequently, our focus in Appendix B.3.2 is on the finetuning approach.

---

> > > > ### Author Response · Authors · 2023-08-16
> > > > **Response Part 2**
> > > >
> > > > > The proposed method does not only decreases the performance (which might be okay if the gap is not significant), but also harms confidence calibration, measured by ECE. It seems authors wanted to claim that the proposed method exhibits better uncertainty by the conformal set size comparison. In my opinion, authors had to focus more on discussion on the discrepancy between ECE and the conformal set size comparison. At this point, ECE is the only reliable metric to me.
> > > >
> > > > In addition to points made in *Calibration Gap* section of our global rebuttal, we would highlight here that no decision in the anytime setting is based on model (or prediction) confidence, i.e. $\max_y p_m(y | \boldsymbol{x})$. This is in contrast to budgeted batch classification setting, where the decision on when to exit could be based on the model confidence. Thus, we believe that our PA potentially undermining model confidence, as evident by poor ECE in the earlier exits, is not a limitation that would seriously impact the merit of our solution. Especially since we focus solely on the anytime setting, where, we would argue, non-monotone behaviour in terms of performance quality is a much more serious issue.
> > > >
> > > > However, it is true that model/prediction confidence, i.e. $\max_y p_m(y | \boldsymbol{x})$, could be useful in anytime scenario as a proxy for model uncertainty. This allows the model, when prompted by its environment to make a prediction, to also offer an estimate of uncertainty alongside that prediction. However, given that our PA potentially adversely affects model confidence in earlier exits, we caution against its use. Instead, we recommend alternative measures of uncertainty that better align with our model, such as the conformal set size. We recognize that this point might not have been explained clearly enough in Sections 6.2-6.3, and we commit to clarifying this aspect in the camera-ready version.
> > > >
> > > > Moreover, we would point out that while the baseline model, such as MSDNet, does display superior model confidence in terms of ECE compared to our PA, its confidence is far from flawless. Specifically, there are data points where the model's confidence trajectory is non-monotone. This means the model starts with high confidence in the initial exits but becomes less certain in subsequent stages. Such a trend is problematic in the anytime setting, similar to the the issues of non-monotonicity in performance quality. It suggests that the model expends additional computational resources only to diminish its confidence in its response. Reassuringly, our PA aids in ensuring monotonicity not only in terms of performance quality (e.g., ground-truth probability) but also in uncertainty aspects (e.g., model confidence or conformal set size, as discussed in Appendix B.7).
> > > >
> > > > Lastly, we would like to highlight that ECE is an imperfect metric with many well-understood flaws. Some common criticisms include
> > > >
> > > >  - difficulties in application to multiclass classification, the equally spaced binning scheme, and the requirement to select an appropriate number of bins [1],
> > > >  - "ECE does not correspond to useful calibration" [2], and
> > > >   - "estimating ECE accurately is difficult because estimators can be strongly biased" [3],
> > > >
> > > > among other criticisms [4, 5, 6]. We acknowledge that ECE is a common metric that can be useful, but we do not see it as "the only reliable metric", or even a particularly reliable metric at all. For this reason, we have used a range of calibration metrics, including ECE, conformal set size, and OC AUC, as together, these provide a holistic view of our method's calibration.
> > > >
> > > >
> > > >
> > > > [1] Jeremy Nixon, Mike Dusenberry, Linchuan Zhang, Ghassen Jerfel, Dustin Tran: Measuring Calibration in Deep Learning. CoRR abs/1904.01685 (2019)
> > > >
> > > > [2] Chenglei Si, Chen Zhao, Sewon Min, Jordan L. Boyd-Graber:
> > > > Re-Examining Calibration: The Case of Question Answering. EMNLP (Findings) 2022: 2814-2829
> > > >
> > > > [3] Matthias Minderer, Josip Djolonga, Rob Romijnders, Frances Hubis, Xiaohua Zhai, Neil Houlsby, Dustin Tran, Mario Lucic:
> > > > Revisiting the Calibration of Modern Neural Networks. NeurIPS 2021: 15682-15694
> > > >
> > > > [4] Rebecca Roelofs, Nicholas Cain, Jonathon Shlens, Michael C. Mozer:
> > > > Mitigating Bias in Calibration Error Estimation. AISTATS 2022: 4036-4054
> > > >
> > > > [5] Juozas Vaicenavicius, David Widmann, Carl R. Andersson, Fredrik Lindsten, Jacob Roll, Thomas B. Schön: Evaluating model calibration in classification. AISTATS 2019: 3459-3467
> > > >
> > > > [6] Kartik Gupta, Amir Rahimi, Thalaiyasingam Ajanthan, Thomas Mensink, Cristian Sminchisescu, Richard Hartley: Calibration of Neural Networks using Splines. ICLR 2021

---

> > > > > ### Author Response · Authors · 2023-08-16
> > > > > **Response Part 3**
> > > > >
> > > > > > (additional question 1)
> > > > >
> > > > > We agree that PA-finetune isn't a universal solution for the ECE gap, and we hope we were transparent enough about this in our work - let us know if you think otherwise. However, we still find it very encouraging that it closes the ECE gap fully on CIFAR datasets, and leads to significant improvements on ImageNet. We take this as an evidence that continuing down this path, by further stabilizing PA finetuning/training, is a promising research direction. However, given that the focus of our work is neither on model confidence/ECE nor on PA finetune, we have not pursued this direction further ourselves. Our central goal is on providing a simple, post-hoc solution that can be applied to any EENN to make it more applicable, i.e. monotone, in the anytime scenario.
> > > > >
> > > > > > I think early-exit scenario is meaningful mostly when we meet a large-scale problem
> > > > >
> > > > > Here we assume that by large-scale problem you mean a problem with a large number of classes. If not let us know and we’ll be happy to respond further.
> > > > >
> > > > > It is not immediately clear to us why would an anytime model be more meaningful on a task with larger number of classes. We believe that an anytime model, which can reliably make use of additional computational resources for enhanced performance, holds equal merit for tasks with fewer classes (like CIFAR-10) as it does for those with a broader class range (such as ImageNet).
> > > > >
> > > > > > (additional question 2)
> > > > >
> > > > > Apologies for the confusion. Our plot could have been made more clear. The blue and purple solid lines overlap in Figure R.1, indicating that the temperature scaling does not affect accuracy for the non-cached version.
> > > > >
> > > > >  For the cached version, where we update the model's prediction at the current exit only if the model confidence surpasses that of previous layers (as detailed in Section 4.1), it is expected that the temperature scaling affects the accuracy. That’s because temperature scaling modifies model confidence which in turn has an effect on how the predictions are cached. This is the reason for (tiny) discrepancy between dashed purple and blue lines in R.1.
> > > > >
> > > > > We hope we have clarified your concerns. If that's the case, we would appreciate if you would consider revising your score. We'd like to respectfully reiterate that most of the issues you brought up pertain to ECE and PA-finetune, both of which are not part of the main message of our work. And in case there are any remaining unaddressed concerns, we are happy to provide further clarifications.

---

> > > > > > ### Comment · Reviewer_6rYn · 2023-08-18
> > > > > > **Thanks for the response**
> > > > > >
> > > > > > Thank you for your response. There is a lot of discussion here, but the major questions remaining for my final evaluation would be whether
> > > > > >
> > > > > > 1. the experimental results support the authors' claim, i.e., whether the performance quality (measured by "max y* prob.") is really improved by the proposed method (especially in the large-scale setting) and
> > > > > >
> > > > > > 2. the proposed method is scalable.
> > > > > >
> > > > > > Below I also leave some detailed comments.
> > > > > >
> > > > > > >> Providing an example scenario with specified numbers would be helpful to understand what it means […]
> > > > > >
> > > > > > > We tried to do so in lines 153-156.
> > > > > >
> > > > > > You can draw a 7-layer NN, specifying y* probs, and how "max y* prob. decrease" is computed. With a similar example, you can also explain how the proposed post-hoc methods perform. This would provide much better understanding of what is going on.
> > > > > >
> > > > > >
> > > > > > >> It would also be nice if there is a metric that quantifies conditional monotonicity by a single number […]
> > > > > >
> > > > > > AUC sounds not optimal but okay to me. Thank you for your consideration.
> > > > > >
> > > > > >
> > > > > > >> could you clarify if you want to improve the model confidence or performance over layers? […]
> > > > > >
> > > > > > I understand that authors want to maximize the probability of the ground-truth class p(y*|x). However, I could not find any experimental result to see if the proposed methods indeed improve it, except for toy examples with only 10 samples like Figure 4 left. Can you evaluate compared method by the average y* probability on CIFAR/ImageNet datasets, maybe drawing a graph of "avg y* prob vs. layer #" and "the AUC of the graph" or "the average of average y* probability per layer"? I feel that this should be aligned with what this work supposed to achieve. Or, please let me know if you already provided the result but I missed it.
> > > > > >
> > > > > >
> > > > > > >> […] "PA finetune" overall decreases the test accuracy, so I wonder if PA is a well-defined learning objective […] random learning objective [...] improved conditional monotonicity or ECE might come from the random noise on the learning objective. […]
> > > > > >
> > > > > > If training from scratch with the "PA finetune" objective converges, then it may not be a random learning objective. I suggest to provide the result, at least in the appendix. In addition, any analysis on its behavior during optimization would be welcome.
> > > > > >
> > > > > > >> ECE is the only reliable metric to me.
> > > > > >
> > > > > > > we would like to highlight that ECE is an imperfect metric with many well-understood flaws.
> > > > > >
> > > > > > In short, to my understanding, authors want to claim that improving ECE is not their main purpose (which is consist with answers to my other questions) and ECE is an imperfect metric. I think the latter is debatable. If authors really want to claim ECE is an imperfect metric, then I think authors should gather all ECE results in a section and discuss the results together with their opinion on ECE, rather than admitting that decreasing ECE is the limitation of the proposed method.
> > > > > >
> > > > > > >> (additional question 1) [...] I think early-exit scenario is meaningful mostly when we meet a large-scale problem
> > > > > >
> > > > > > Large-scale here implies both the resolution of the images and the number of classes, however, the resolution would be a more important factor when considering the computational cost. Though the CIFAR datasets has been widely used to quickly evaluate ideas, the results are not often scalable, such that it is not usable in real world applications.
> > > > > >
> > > > > > To make sure, I raised a concern on the scalability because ECE becomes worse when PA or PA-finetune is applied on ImageNet, which is the only large-scale image dataset throughout experiments. Maybe authors want to try other large-scale datasets, e.g., place365 or iNaturalist dataset, to see if ImageNet is the exceptional case. Or, authors may want to argue the imperfectness of ECE.

---

> > > > > > > ### Author Response · Authors · 2023-08-18
> > > > > > > **Response 2 Part 1**
> > > > > > >
> > > > > > > We thank you for your continuing engagement during the discussion period, we appreciate it a lot!
> > > > > > >
> > > > > > > > 1. the experimental results support the authors' claim, i.e., whether the performance quality (measured by "max y* prob.") is really improved by the proposed method (especially in the large-scale setting)
> > > > > > >
> > > > > > > We are encouraged to see that your previous concerns related to whether our focus is on performance quality or prediction confidence seem to be resolved.
> > > > > > >
> > > > > > > Our central claim in the paper is that *our method leads to models that are more monotone over exits in the performance quality* as measured by e.g. ground-truth probability. This is paramount in the anytime setting where we desire models that can use additional compute to further refine their predictions. We illustrate this in Figure 3 - there we see that state-of-the-art EENNs, e.g. MSDNet or DViT, have test points that are non-monotone in performance quality, and that our PA leads to large improvements in this regard, i.e. the % of test points with a decrease in ground-truth probability is significantly smaller.
> > > > > > >
> > > > > > > Here we would also refer to reviewer kJKM that wrote: *The provided evaluations are comprehensive and support the claims made in the paper.*
> > > > > > >
> > > > > > > > I understand that authors want to maximize the probability of the ground-truth class p(y*|x). […]
> > > > > > >
> > > > > > > As aforementioned, our primary goal is to *achieve monotonicity in performance quality*, e.g. ground-truth probability, which is not equivalent to maximizing performance quality overall. In our opinion, expecting a post-hoc method like ours to significantly surpass the average performance quality of the baseline model would be overly optimistic. However, a more realistic goal is to fix a pathology of the baseline models that exhibits the non-monotone behavior in the performance quality across exits. This is precisely what our model accomplishes, as depicted in Figure 3. Importantly, we attain this without sacrificing the average performance quality, as evidenced in Figure 2.
> > > > > > >
> > > > > > > > 2. the proposed method is scalable.
> > > > > > >
> > > > > > > Note that our method is entirely post-hoc, so it incurs no computational overhead over the backbone EENN, e.g. MSDNet. As such, our method maintains the scalability of state-of-the-art EENNs when it comes to image resolution/size.
> > > > > > >
> > > > > > > Recall that our main goal is introducing conditional monotonicity of performance quality, e.g. ground-truth probability. So when it comes to number of classes, the benefit of our method actually increases on task with larger number of classes.  As seen in Figure 3, the monotonicity of our method is the best for ImageNet where we achieve perfect conditional monotonicity, i.e. for no test examples we observe any drop in the ground-truth probability as more exits are evaluated. We find this very encouraging, as we agree with you that many relevant tasks consists of large number of classes.
> > > > > > >
> > > > > > > > ImageNet, which is the only large-scale image dataset throughout experiments. Maybe authors want to try other large-scale datasets, e.g., place365 or iNaturalist dataset, to see if ImageNet is the exceptional case
> > > > > > >
> > > > > > > Even in the most recent and state-of-the-art EENN papers that we are aware of, see DViT [1] and L2W [2], ImageNet is the largest image dataset considered. If you know of any EENN model that considers some of the datasets you mention, please point it out to us and we will gladly try out our proposed PA on it (which should be easy given the PA's post-hoc nature).

---

> > > > > > > > ### Author Response · Authors · 2023-08-18
> > > > > > > > **Response 2 Part 2**
> > > > > > > >
> > > > > > > > > To make sure, I raised a concern on the scalability because ECE becomes worse when PA or PA-finetune is applied on ImageNet
> > > > > > > >
> > > > > > > > It is true that ECE of our method is worse on ImageNet as compared to CIFAR datasets. However, as mentioned in our previous response, ECE measures the quality of model/prediction confidence. Since no decision in our considered setting (anytime) is based on prediction confidence, we would politely repeat that we do not see this as a big detriment of our model.
> > > > > > > >
> > > > > > > > > If authors really want to claim ECE is an imperfect metric, then I think authors should gather all ECE results in a section and discuss the results together with their opinion on ECE, rather than admitting that decreasing ECE is the limitation of the proposed method.
> > > > > > > >
> > > > > > > > We included ECE results in the limitations section primarily due to its widespread use, and because it allowed us to motivate approaches like PA with adaptive thresholding and PA-finetune which we believe present a promising direction for future research of deep anytime models. That said, we recognize that discussing ECE in the limitations section, without adequately explaining its lesser relevance in the anytime setting, might lead to confusion. Our discussion with you helped us greatly in realising this, for which we would like to thank you.
> > > > > > > >
> > > > > > > > We will make sure to correct this confusion in the camera-ready. Concretely, we will add to Appendix B.3 a summary of our previous response to you (c.f. Response Part 2) where we tried to explain in detail why ECE is of lesser importance in the anytime setting. In Appendix B.3.2, we also already present ECE results for all datasets using MSDNet backbone. Let us know if you think this is sufficient, or if you believe we need to include any further explanations/experiments.
> > > > > > > >
> > > > > > > > >  I suggest to provide the result, at least in the appendix. In addition, any analysis on its behavior during optimization would be welcome.
> > > > > > > >
> > > > > > > > Thanks for the suggestion, we will add PA-train-from-scratch results to Appendix and describe the issues that arise during training to provide more pointers for future research. Note that Product-of-Experts objectives are known to be tricky to work with during training, see for example [3].
> > > > > > > >
> > > > > > > > As a preview, we attach results for MSDNet on CIFAR-100 dataset below here
> > > > > > > >
> > > > > > > > **Test accuracy [%]**
> > > > > > > >
> > > > > > > > |  | 1 | 2 | 3 | 4 | 5 | 6 | 7 |
> > > > > > > > | --- | --- | --- | --- | --- | --- | --- | --- |
> > > > > > > > | PA-finetune | 63.5 | 67.6 | 70.0 | 71.5 | 72.3 | 73.1 | 73.7 |
> > > > > > > > | PA-train | 61.9 | 64.2 | 65.7 | 66.0 | 66.6 | 67.1 | 67.3 |
> > > > > > > >
> > > > > > > > Convergence of PA-train model is to us a clear indication that PA is not equivalent to 'random learning objective'.
> > > > > > > >
> > > > > > > > > AUC sounds not optimal but okay to me. Thank you for your consideration.
> > > > > > > >
> > > > > > > > We agree with you that AUC might not be optimal. After your suggestion in your previous response, we gave this further thought and we realized that an Expected Maximum Decrease  in ground-truth probability (EMD) might be an even better metric
> > > > > > > >
> > > > > > > > $$
> > > > > > > > EDM := \sum_{d \in [0, 1]} d \cdot p(d)
> > > > > > > > $$
> > > > > > > >
> > > > > > > > where $d$ is a maximum decrease and $p(d)$ is the proportion of test points that have a maximum decrease of $d$. In practice, we approximate $[0, 1]$ with a grid, e.g. `np.linspace(0, 1, B)` where B represents the number of points on the grid.
> > > > > > > >
> > > > > > > > Below we attach results for MSDNet on CIFAR-100  (using B=15)
> > > > > > > >
> > > > > > > > |  | baseline | PA | CA |
> > > > > > > > | --- | --- | --- | --- |
> > > > > > > > | EDM | 0.176 | **0.002** | 0.047 |
> > > > > > > >
> > > > > > > > Let us know if there are any other unaddressed concerns. We would be very glad to discuss them before the end of the discussion period.
> > > > > > > >
> > > > > > > > [1] Wang, Y., et al., 2021. Not All Images are Worth 16x16 Words: Dynamic Transformers for Efficient Image Recognition. *NeurIPS*
> > > > > > > >
> > > > > > > > [2] Han, Y., et al., 2022. Learning to Weight Samples for Dynamic Early-exiting Networks. *ECCV*
> > > > > > > >
> > > > > > > > [3] Hinton, Geoffrey, 2000. Training Products of Experts by Minimizing Contrastive Divergence

---

### Author Rebuttal · Authors · 2023-08-08

We thank all reviewers for their valuable time and feedback; it’s much appreciated.

We are encouraged that the reviewers found “the paper well written and easy to follow” (G2Sm), thought that “the studied problem is interesting and important” (NWJy), “often overlooked by the relevant literature” (ne1V), and that we  “tackle a practical and important problem that should be solved” (kJKM). Moreover, we are pleased to hear that our proposed method is “well motivated and designed” (G2Sm) and that “its effectiveness is also supported by theoretical results” (6rYn). Importantly, reviewers found the experiments “comprehensive” (NWJy & G2Sm & kJKM), “supporting the claims made in the paper” (kJKM), and “wide and insightful” (ne1V). Finally, we are excited that the reviewers recognized that our work “can motivate further research in the topic” (ne1v) and “could aid/inspire future research” (G2Sm).

We next address some of the main concerns raised in the reviews.

**Anytime Prediction vs. Input-Dependent/Efficient Inference setting** (kJKM, ne1V, NWJy)

We make no claims in our paper regarding the use of our model in an input-dependent/efficient inference setting (a.k.a. budgeted batch classification [1]). **Our focus is solely on the anytime scenario**, where the computational budget is unknown or dynamic, and exiting is determined by the environment (cf. “Setting of Interest”, lines 72-81). Since our solution is very lightweight (4 lines of Python code) and can be applied post-hoc, it is easy to “turn on and off”. Hence, the same early-exit model, e.g., MSDNet, can easily be used for either scenario as required.

One of our main goals is to highlight the distinction between the two settings and their different requirements.  We believe that this is missing in the current literature where the desiderata of anytime setting have been largely overlooked (as evidenced by the highly non-monotone behavior of SOTA ‘anytime’ models, cf. Figure 1)

Reviewer NWJy has suggested that it would be interesting to study the effect of monotonicity in the budgeted batch classification scenario too. This is a great idea, but it is out of the scope of the current paper.  We plan to investigate it in future work.

**Monotonicity via Calibration** (6rYN, kJKM, G2Sm, ne1V)

Some reviewers questioned whether monotonicity could be achieved by calibrating the underlying early-exit NN, rather than employing our proposed PA.  To answer this, we conducted a new experiment where we calibrated MSDNet and analyzed its effect on monotonicity. For the calibration, we utilized three standard techniques: temperature scaling [2], deep ensembles  [3], and last-layer Laplace [4].

In Figure R.1 (see attached pdf) we present results for MSDNet on CIFAR-100. While all three approaches lead to better ECE (right plot), our PA significantly outperforms them in conditional monotonicity (middle plot). Moreover, all three baselines are (arguably) more complex compared to our PA: temperature scaling requires validation data, deep ensemble has M-times more parameters (M = ensemble size), and Laplace is significantly slower compared to PA since we need test-time sampling to estimate the predictive posterior at each exit.

However, simply improving calibration might not be expected to improve monotonicity. Thus, in Figure R.1, we also provide results for combining the above uncertainty calibration techniques with our CA baseline, as suggested by reviewer 6rYn. We see that this combination does indeed provide further improvements w.r.t. monotonicity – the improved calibration allows for better caching of the correct prediction. **However, the monotonicity of these caching-and-calibrated methods still underperforms compared to PA, despite being more complicated**.

We plan to include these results in our paper's camera-ready version, as it neatly demonstrates that monotonicity can not be achieved via calibration alone. This insight further underlines the value of our proposed approach. We thank the reviewers for this suggestion.

**Calibration Gap** (6rYN, kJKM, ne1V)

Some reviewers raised concerns that our post-hoc method can hurt ECE in earlier exits (c.f. Figure 5). While we share this concern, we would like to emphasize that this potential drawback must be considered alongside the notable advantages of our method. These include the introduction of conditional monotonicity and enhanced uncertainty estimates (c.f. Figure 4, right). Additionally, note that poor ECE early on stems from under-confidence rather than over-confidence. We would argue such inductive bias is appropriate in EENNs, especially in safety-critical scenarios

**Moreover, in addition to pointing out this limitation ourselves, we propose two concrete ways to improve the calibration in earlier exits in the paper**, one based on the training/finetuning with the Product Anytime (PA) objective and one based on using ReLU with adaptive thresholding. We describe both in detail and present their impact on the monotonicity/calibration in Appendix B.3. Both yield very promising results when it comes to closing the calibration gap, albeit at some cost to monotonicity/accuracy. We will also include a description of both approaches in the main text using the additional page in the camera-ready.

Lastly, we believe that our paper is stronger due to our honest discussion of both the pros and cons of the method, which we hope will facilitate future work and easy adoption for practitioners.

We further address all reviewers' remaining concerns and questions in individual responses.

[1] Huang, G., et al., 2017. Multi-scale dense networks for resource efficient image classification. *ICLR*

[2] Guo, C., et al., 2017, July. On calibration of modern neural networks. *ICML*

[3] Lakshminarayanan, B., et al., 2017. Simple and scalable predictive uncertainty estimation using deep ensembles. *NeurIPS*

[4] Daxberger, E., et al.., 2021. Laplace redux-effortless bayesian deep learning. *NeurIPS*

---

### Decision · Program_Chairs · 2023-09-21

**Decision:**

Accept (poster)

**Comment:**

This paper studies conditional monotonicity for anytime prediction models, where the predicted solution should monotonically improve (i.e. the confidence in the correct answer) through the network. This potentially allows a setting where a "best effort" prediction is made if the prediction the compute is interrupted, or if the same model needs to be deployed in many devices with different capacities. The authors focus on the compute vision domain and MSDNet and first demonstrate that a constrained-free model doesn't satisfy their desired conditional monotonicity. They then propose a solution. First, they discuss a simple solution for guaranteeing monotonicity, but can hurt accuracy. Instead, they propose a relaxed version that still promotes monotonicity and empirically leads to comparable accuracy.

The rebuttal and discussion period were very active with many great interactions and suggestions that the authors stated they will include in the paper. Most reviewers were convinced that the contributions of this work and the extensive analysis and insights are meaningful. The main remaining concerns were around the importance and practicality of monotonicity (kJKM, 6rYn) and around the reduction in ECE (6rYn). While it would have definitely strengthen the paper if the authors could have used their method for obtaining practical gains in instance-wise confidence-based early exiting, studying monotonicity by itself can also be valuable. Also, the comprehensive analysis of the authors which includes also some negative results such as increased ECE should be encouraged. Furthermore, the improved results on other confidence-dependent metrics such as conformal set size are stronger since they rely on stronger finite-set guarantees and more stable than ECE.

The authors should incorporate all the new results and comments from the discussion period into the paper. The scope of the paper should also be more properly defined in order to avoid confusion and false expectations. I would suggest to limit the discussion on early exit models since many recent paper use this term for describing confidence-based exit functionalities. Instead, anytime prediction describes better the examined setting where the the halting criterion is based on the environment instead of the specific instance. This should also alleviate some of the concerns around the practicality of the monotonicity requirement which might or might not be helpful for early exiting, but is more clearly helpful for anytime predictors.